RESEARCH COMMUNICATION

# Spliceosome factors target timeless (*tim*) mRNA to control clock protein accumulation and circadian behavior in Drosophila

Iryna Shakhmantsir[1], Soumyashant Nayak[2,3], Gregory R Grant[2,3], Amita Sehgal[1,2,3]*

[1]Chronobiology Program at Penn, Howard Hughes Medical Institute, Perelman School of Medicine at the University of Pennsylvania, Philadelphia, United States; [2]The Institute for Translational Medicine and Therapeutics, Perelman School of Medicine at the University of Pennsylvania, Philadelphia, United States; [3]Department of Genetics, Perelman School of Medicine at the University of Pennsylvania, Philadelphia, United States

*For correspondence:
amita@pennmedicine.upenn.edu

**Competing interests:** The authors declare that no competing interests exist.

**Abstract** Transcription-translation feedback loops that comprise eukaryotic circadian clocks rely upon temporal delays that separate the phase of active transcription of clock genes, such as Drosophila *period (per)* and *timeless (tim),* from negative feedback by the two proteins. However, our understanding of the mechanisms involved is incomplete. Through an RNA interference screen, we found that pre-mRNA processing 4 (PRP4) kinase, a component of the U4/U5.U6 triple small nuclear ribonucleoprotein (tri-snRNP) spliceosome, and other tri-snRNP components regulate cycling of the molecular clock as well as rest:activity rhythms. Unbiased RNA-Sequencing uncovered an alternatively spliced intron in *tim* whose increased retention upon *prp4* downregulation leads to decreased TIM levels. We demonstrate that the splicing of *tim* is rhythmic with a phase that parallels delayed accumulation of the protein in a 24 hr cycle. We propose that alternative splicing constitutes an important clock mechanism for delaying the daily accumulation of clock proteins, and thereby negative feedback by them.

**Editorial note:** This article has been through an editorial process in which the authors decide how to respond to the issues raised during peer review. The Reviewing Editor's assessment is that all the issues have been addressed (see decision letter).

DOI: https://doi.org/10.7554/eLife.39821.001

## Introduction

Circadian rhythms allow organisms to orchestrate behavioral and physiological outputs in anticipation of predictable diurnal changes in the environment. These rhythms are generated by endogenous molecular clocks that entrain to environmental cycles, predominantly light and temperature, and can maintain rhythms when released into constant conditions (free-run). A conserved mechanistic feature of circadian clocks is an auto-regulatory transcriptional feedback loop in which circadian proteins rhythmically regulate their own expression to generate a clock, and also drive a global program of cycling gene expression. Discovery of this clock mechanism was revolutionary and has received well-deserved recognition, but critical aspects of how the clock is sustained remain unclear.

PERIOD (PER) and TIMELESS (TIM) are the auto-regulating elements of the circadian clock in Drosophila (*Allada and Chung, 2010*; *Hardin, 2011*; *Zheng and Sehgal, 2012*). The expression of *per* and *tim* is driven by circadian transcription factors CLOCK (CLK) and CYCLE (CYC), and peaks

around the early night. Relative to their mRNA peak, accumulation of PER and TIM proteins is delayed by ~6 hr. In the mid-to-late night, PER and TIM are predominantly nuclear, and once in the nucleus, they repress CLK-CYC activity to decrease *per* and *tim* expression. Degradation of TIM and then of PER in the morning resets the transcription cycle and restarts the loop. In order to maintain rhythmicity and set the proper pace of the circadian clock, both PER and TIM need to be dynamically regulated on multiple levels. For instance, a stable circadian molecular oscillator requires temporal delays to separate the phases of gene transcription and repression and thereby prevent these from reaching equilibrium (*Zheng and Sehgal, 2012*).

The overall levels and stability of TIM constitute a critical circadian modality. Although PER is the more important factor for transcriptional regulation, levels and activity of PER depend upon TIM (*Price et al., 1995*; *Dubruille and Emery, 2008*). TIM levels are acutely modulated by light, which promotes the degradation of TIM, and thereby PER, during the day and allows the rise in circadian transcription (*Suri et al., 1999*; *Yang and Sehgal, 2001*). Subsequently, TIM accumulation is necessary to stabilize PER and promote its nuclear accumulation (*Jang et al., 2015*). Thus, in the presence of light:dark cycles, light delays the accumulation of TIM-PER and so contributes to the lag in repression. These temporal relationships are largely preserved in constant darkness, and are also entrained by temperature cycles regardless of light cues, although the mechanisms under these conditions are not known.

While regulated protein stability and translation have been directly explored as mechanisms that could contribute to maintenance of the feedback loop (*Dembinska et al., 1997*; *Chen et al., 1998* ; *Lim and Allada, 2013*; *Zhang et al., 2013*), and regulation of protein stability is indeed critical (*Zheng and Sehgal, 2012*), little investigation has focused on a potential role of alternative splicing. To date, the best-studied role for alternative splicing in Drosophila rhythms is in the temperature-dependence of the behavioral siesta (*Majercak et al., 1999*; *Majercak et al., 2004*; *Collins et al., 2004*). Splicing is driven by spliceosomes, dynamic RNA-protein complexes composed of five core small nuclear ribonucleoprotein particles (U1, U2, U4, U5, U6 snRNP) and >150 additional proteins specific for each snRNP complex (*Wahl et al., 2009*). In this study, we report a circadian role for Pre-mRNA Processing factor 4 (PRP4), a conserved component of the spliceosomal U4/U6.U5 triple small nuclear ribonucleoprotein (tri-snRNP) complex. We identified PRP4 in a screen for novel regulators of the free-running circadian period, and established that PRP4 is necessary in *tim+* clock cells to maintain 24 hr period and robust rhythms of the circadian clock. In addition to *prp4,* downregulation of multiple tri-snRNP components affected circadian period length and rhythmicity, which led us to implicate this entire spliceosomal complex in circadian regulation. Using unbiased RNA-Sequencing, we characterized the splicing events regulated by PRP4 and identified a novel intron retention event in *tim.* We show that alternative splicing of this intron in *tim* represents an important mechanism to time the daily accumulation of TIM, in constant darkness following entrainment to light:dark cycles and also in temperature cycles. Together, these findings identify a mechanism contributing to the maintenance of clock function.

## Results

### Pre-mRNA splicing factor four is a new regulator of the circadian clock

In a screen for kinases that affect circadian period length of Drosophila rest:activity rhythms in constant dark:dark (DD) conditions, we identified Pre-mRNA Processing factor 4 (PRP4), a splicing factor that also has kinase activity (*Kojima et al., 2001*; *Schneider et al., 2010*). RNA interference (RNAi)-mediated knockdown of *prp4* in *tim+* clock neurons with the *Tim-UAS-Gal4* (*TUG*) Gal4 driver resulted in consistently long free-running circadian periods (*Figure 1A–B*). Two independent RNAi lines (GD and KK) were used to confirm these findings.

The length of the free-running circadian period is an important parameter that also affects the daily distribution of fly activity in light:dark (LD) cycles. Thus, we determined the effect of *prp4* knockdown on fly diurnal activity in LD conditions (*Figure 1C*). Control (*TUG; Dcr2/+*) flies displayed characteristic bimodal activity in LD, with activity peaks that precede the onset of light (ZT0) and the onset of dark (ZT12). Compared to controls, the flies with circadian-cell-specific *prp4* knockdown (*TUG; Dcr2 >prp4^RNAi^(GD)*) exhibited a delayed evening activity peak as well as a slightly delayed

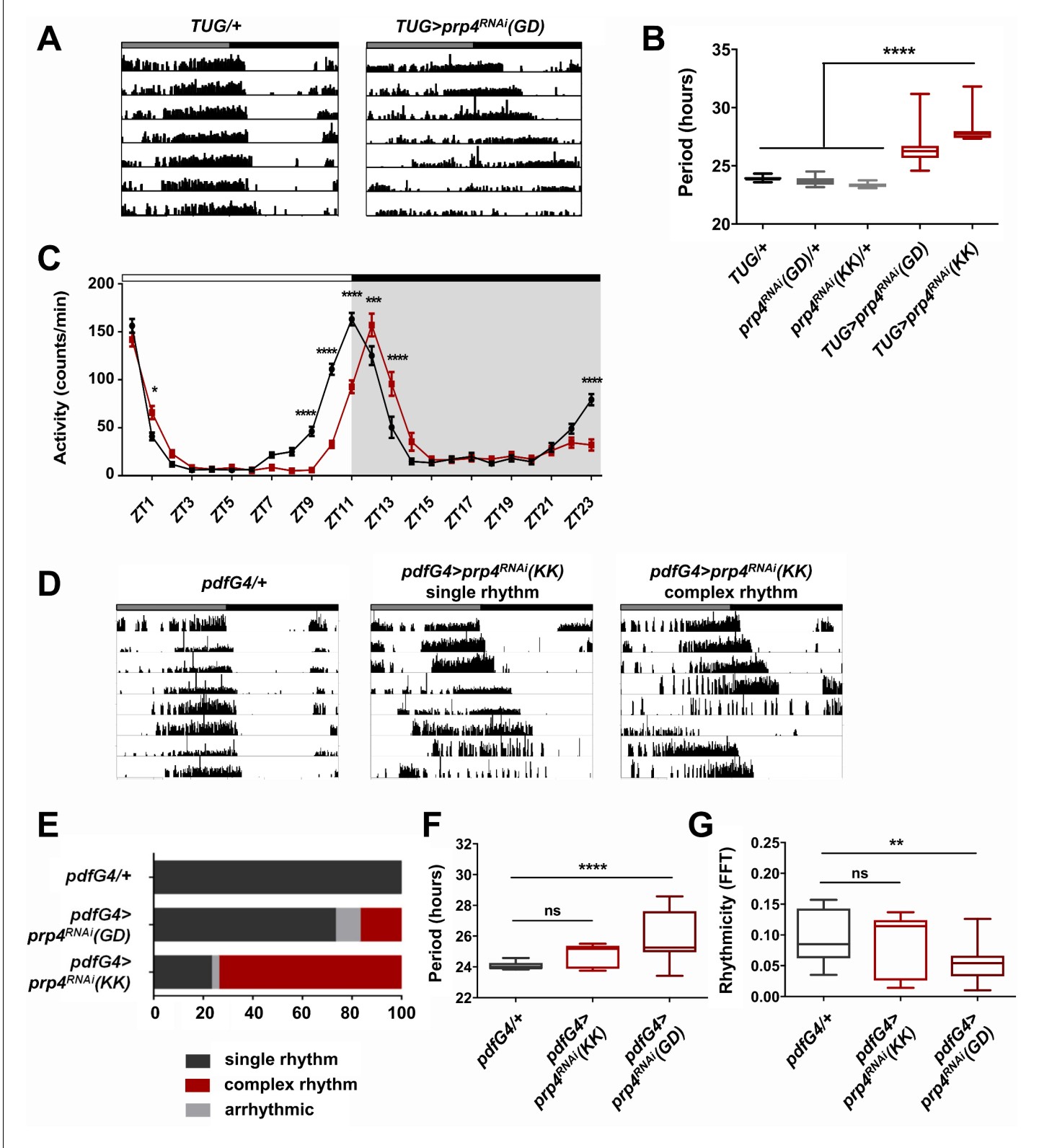

**Figure 1.** Pre-mRNA splicing factor four is a new regulator of the circadian clock. (**A**) Representative activity records of free-running fly behavior upon downregulation of *prp4* in *tim+* neurons. *Dicer2* (*Dcr2*) was co-expressed with the RNAi transgenes to increase the knockdown efficiency. Genotypes are indicated on top of each panel. The gray and black bars indicate the subjective day and night, respectively. (**B**) The lengthening of free-running circadian period is significant for two independent *prp4* RNAi lines (GD and KK). ****p ≤ 0.0001 relative to heterozygous controls by one-way ANOVA

*Figure 1 continued*

and Tukey's post hoc test, n = 8–24. (**C**) Downregulation of *prp4* in *tim+* cells affects morning and evening anticipation in 12 hr:12 hr light:dark (LD) conditions. The activity profile of control (*TUG; Dcr2/+*) flies is in black, while the activity profile of experimental (*TUG; Dcr2 >prp4^RNAi^(GD)*) flies is in red. The white and black bars indicate light and dark conditions, respectively. *p ≤ 0.05, ***p ≤ 0.001,****p ≤ 0.0001 to control (*TUG; Dcr2/+*) for each ZT range by two-way ANOVA and Sidak's post hoc test. Data represent mean ±SEM (n = 31–32). (**D**) Activity records demonstrate 7 days of representative free-running rhythms of *prp4* knockdown flies. *Dicer2 (Dcr2)* was co-expressed with the RNAi transgenes to increase the knockdown efficiency. Genotypes are indicated on top of each panel. The gray and black bars indicate the subjective day and night, respectively. (**E**) Knockdown of *prp4* in LNv pacemaker cells causes complex behavioral periods (p < 0.0001 by $\chi^2$ analysis, n = 19–38). (**F, G**) Knockdown of *prp4* in LNvs lengthens circadian period and decreases rhythm strength. n.s., not significant at the 0.05 level, **p ≤ 0.01, ****p ≤ 0.0001 to control (*pdfG4; Dcr2/+*) by one-way ANOVA and Holm-Sidak's post hoc test. Only rhythmic flies (FFT > 0.01) were analyzed (n = 9–22). In panels (**B**), (**F**) and (**G**), the boxes extend from the 25th to 75th percentiles, the line within the box is plotted at the median and whiskers extend from the lowest to highest value.

DOI: https://doi.org/10.7554/eLife.39821.002

morning peak. These findings are consistent with the longer endogenous period that we report for *prp4* knockdown flies (*Figure 1A–B*).

As the small lateral ventral neurons (s-LNvs) are the most relevant clock cells for driving rest:activity rhythms under constant dark conditions, we asked if effects of PRP4 were mediated in these cells (*Helfrich-Förster, 1998*). To test this hypothesis, we downregulated *prp4* in both s-LNvs and large lateral ventral neurons (l-LNvs) with a *Gal4* transgene driven by a peptide expressed specifically in these cells, Pigment Dispersing Factor (PDF). Knockdown of *prp4* with *pdf-Gal4* driving the weaker RNAi line (GD) led to a modest yet significant lengthening of free-running circadian period (*Figure 1F*). Interestingly, the knockdown of *prp4* with a stronger RNAi (KK) led to complex rhythms (*Figure 1E*). These complex rhythms were generally characterized by changes in the behavioral pattern, often manifest as phase shifts during day 4 or 5 of constant darkness (see ~6 hr shift in the record shown), in the midst of an otherwise rhythmic record (*Figure 1D*). Such a complex rhythm phenotype could reflect uncoupling of period into two components of *pdf+* and *pdf-* cell oscillators (*Yao et al., 2016*). Overall, these findings suggest that PRP4 action in LNvs contributes to maintaining clock function.

## PRP4 is required for robust TIM and PER cycling

Since our data indicated that PRP4 is necessary in LNvs for proper circadian rhythmicity (*Figure 1*), we next examined the circadian cycling of PER and TIM in s-LNvs at regular intervals around the clock (*Figure 2A–C*). Cycling of both PER and TIM was affected in s-LNvs upon *prp4* knockdown with the clock-specific *TUG* driver (*Figure 2A–C*). Total TIM levels, as quantified by corrected total cell fluorescence (CTCF) image analysis, were decreased during the night (*Figure 2B–C*). PER levels were also lower in the late night (ZT20), and especially so in the early morning (ZT2), in *prp4* knockdown flies (*Figure 2A,C*). Additionally, nuclear accumulation of PER was delayed beyond ~ZT20, the time point at which PER was already partially localized to the nucleus in control flies (*Figure 2A*). To further characterize the effect on nuclear expression, we profiled relative nuclear PER expression upon *prp4* knockdown with each of the two RNAi lines (GD and KK) in s-LNvs around the time of nuclear accumulation of PER (ZT18-ZT22). We found that the nuclear accumulation of PER was slower in flies with decreased *prp4* (both GD and KK RNAi lines) than in controls (*TUG; Dcr2/+*) at these time points (*Figure 2—figure supplement 1*). Overall, our immunohistochemistry data point to a distinct molecular clock phenotype upon *prp4* downregulation. Because TIM is necessary for nuclear accumulation of PER (*Vosshall et al., 1994*; *Saez and Young, 1996*; *Jang et al., 2015*), reduced TIM accumulation could account for the delay and overall reduced nuclear expression of PER in s-LNvs (*Figure 2*, *Figure 2—figure supplement 1*), which together could account for the longer free-running period.

As biochemical analysis of PER and TIM oscillations requires large amounts of tissue, we conducted pan-neuronal knockdown of *prp4* with the *elavGal4* driver. The pan-neuronal manipulation targeted *prp4* more broadly, allowing us to verify the efficiency of the knockdown through RNA analysis of whole heads. We consistently observed ~50% reduction in *prp4* RNA in head lysates of flies where it was knocked down pan-neuronally (*Figure 2—figure supplement 2*). This broad manipulation also allowed us to test the effects of *prp4* reduction on PER and TIM levels using western blotting of whole head lysates. The results obtained with this relatively crude method agreed

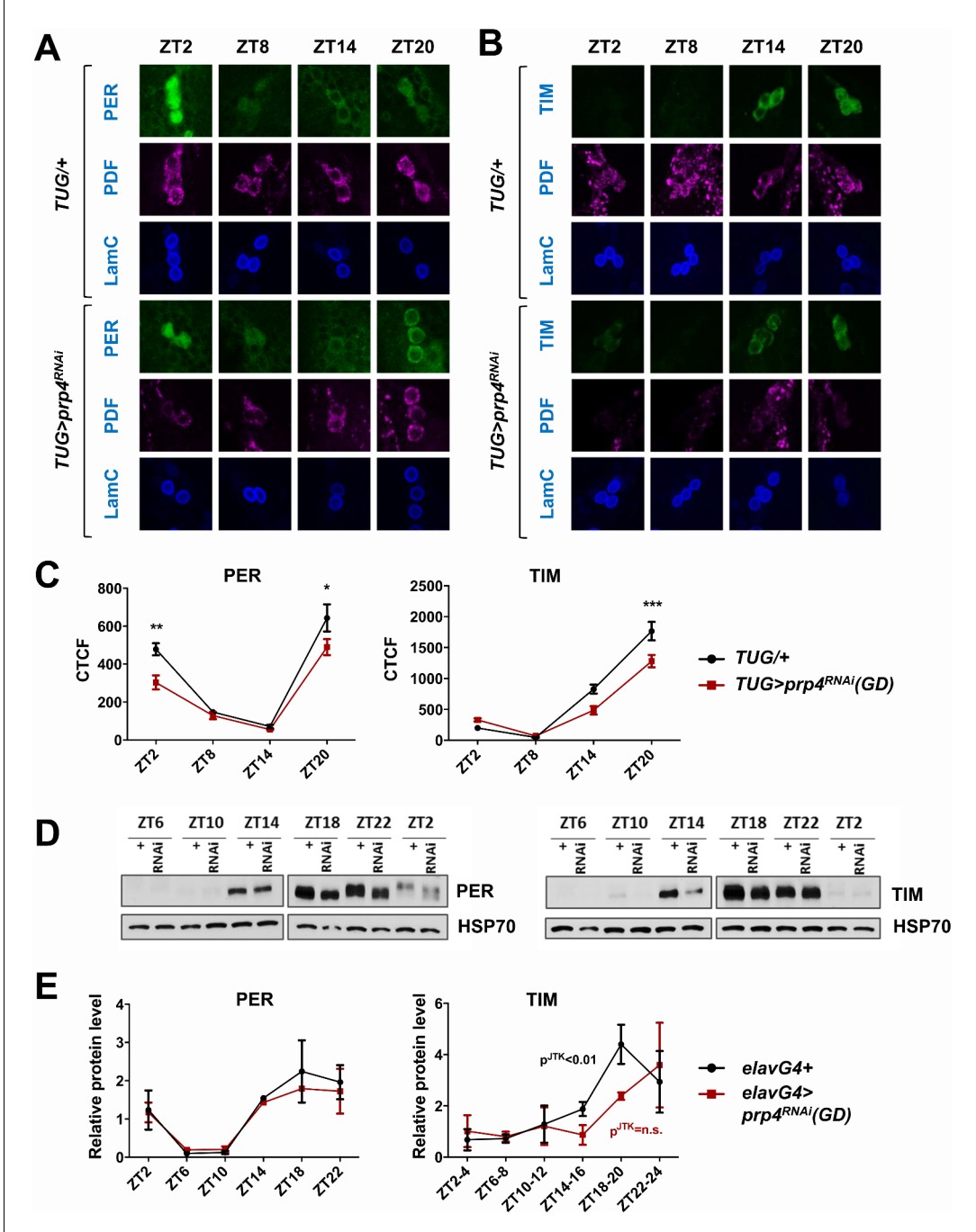

**Figure 2.** PRP4 is required for robust TIM and PER cycling. (A, B) Cycling of PER and TIM is disrupted in s-LNvs of *prp4* knockdown flies. Adult brains were dissected at time points indicated and immunostained with PER or TIM (green), PDF (magenta) and LaminC (blue) antibodies. *Dicer2 (Dcr2)* was co-expressed with the *prp4* RNAi transgene to increase its knockdown efficiency. Genotypes are indicated on the sides of each panel. The displayed images are representative of 2–3 independent experiments. (C) Corrected Total Cell Fluorescence (CTCF) was used to quantitatively assess the change in levels of PER and TIM in s-LNvs. The signal from both the nucleus and the cytoplasm was used to calculate CTCF. 10–21 cells from 5 to 8 brains were analyzed for ZT2, ZT14 and ZT20 and 8–10 cells from 3 to 5 brains for ZT8. Images were taken with identical confocal settings. *$p \leq 0.05$, **$p \leq 0.01$, ***$p \leq 0.001$ to control (*TUG;Dcr2/+*) for each ZT time point as determined by two-way ANOVA and Sidak's post hoc test. Error bars = ± SEM. (D) PER phosphorylation is decreased and cycling of TIM is blunted in fly heads with pan-neuronal knockdown of *prp4*. *Dicer2 (Dcr2)* was co-expressed with the *prp4* RNAi transgene to increase its knockdown efficiency. Adult fly heads were collected at indicated zeitgeber (ZT) time points in a 12 hr:12 hr light: dark cycle. Representative western blots probed for PER (right) or TIM (left) are shown. HSP70 was used as a loading control. (E) Total PER and TIM levels were quantified from western blots in (D). JTK cycle analysis identified significant cycling ($p^{JTK} \leq 0.01$) for control (*elavGal4; Dcr2/+*) and not

*Figure 2 continued on next page*

*Figure 2 continued*

significant (n.s at the 0.05 level) cycling for samples with pan-neuronal *prp4* knockdown. Data represent mean ±SEM (n = 2–3 independent experiments).

DOI: https://doi.org/10.7554/eLife.39821.003

The following figure supplements are available for figure 2:

**Figure supplement 1.** *prp4* downregulation delays night-time accumulation of PER in nuclei of s-LNvs.

DOI: https://doi.org/10.7554/eLife.39821.004

**Figure supplement 2.** *prp4* is efficiently knocked down with the GD RNAi line.

DOI: https://doi.org/10.7554/eLife.39821.005

well with our immunohistochemistry profiling of s-LNvs. First, we observed a strong effect of *prp4* downregulation on total TIM levels, particularly during the initial TIM accumulation phase (ZT10-18) (*Figure 2D–E*). Quantification of our western blot data across multiple experiments suggested that *prp4* knockdown also blunted the circadian cycling of TIM protein (*Figure 2E*). While the total levels of PER were not changed upon *prp4* downregulation, its phosphorylation profile was dampened. In control (*elavGal4; Dcr2/+*) flies, as previously reported (*Edery et al., 1994*), PER was increasingly phosphorylated in the late night and early morning up until its degradation. In flies with reduced *prp4* levels, PER migrated at a lower molecular weight on western blots (ZT18-ZT2). We speculate that the dampened phosphorylation profile of PER reflects, at least in part, the defect in nuclear accumulation of PER because a number of sites on PER get phosphorylated only when PER starts to accumulate in the nucleus (after ZT20) (*Chiu et al., 2011*). Alternatively, *prp4* depletion could have a TIM-independent effect on PER to regulate its phosphorylation profile.

## Circadian rhythmicity is modulated by tri-snRNP levels

As noted above, PRP4 is associated with kinase activity, but it is also a component of the spliceosome (*Kojima et al., 2001*; *Schneider et al., 2010*). To address if the circadian role of PRP4 was spliceosome-dependent, we assayed circadian behavior following knockdown of additional tri-snRNP components. Strikingly, clock-specific downregulation of *pre-mRNA processing factor 3* (*prp3*) and *pre-mRNA processing factor 31* (*prp31*), which are associated primarily with U4/U6 snRNP, as well as *pre-mRNA processing factor 8* (*prp8*) and *bad response to refrigeration 2* (*brr2*), which are associated with U5 snRNP, caused pronounced defects in free-running circadian behavior (*Table 1*). The phenotypes ranged from period lengthening to complete arrhythmicity, suggesting that the overall levels and/or stability of tri-snRNP regulate circadian rhythms.

To determine if the molecular signatures of *prp4* knockdown and other tri-snRNP downregulation phenotypes were similar, we performed s-LNv-specific analysis of PER and TIM levels with a few randomly selected RNAi lines. We found that the s-LNvs of flies with downregulated *prp8* had lower TIM levels at night, and also showed decreased nuclear accumulation at ZT20 relative to control flies (*Figure 3A*). These data allow us to conclude that circadian oscillations of PER (data not shown) and TIM are sensitive to changes in the levels of multiple tri-snRNP components.

We further examined tri-snRNP function in circadian regulation by utilizing some of the previously characterized mutants of *prp8* and *brr2* (*Coelho et al., 2005*; *Bivik et al., 2015*). Because tri-snRNP components are essential genes with no homozygous viable knockout mutants described to date, we tested circadian behavior in the viable transheterozygous *prp8/brr2* mutants. The heterozygous *prp8/+* and *brr2/+* mutants displayed normal circadian rhythms, readily explained by the haplosufficiency of these two genes (*Figure 3B*). Transheterozygous *prp8/brr2* mutant flies had normal circadian period (data not shown), but displayed decreased rhythm strength compared to the *prp8/+* and *brr2/+* heterozygous flies, confirming a circadian function of the tri-snRNP (*Figure 3B*).

## PRP4 regulates *tim* splicing

Next, we attempted to identify the mechanism by which PRP4 regulates circadian rhythms, starting with the broad hypothesis that downregulation of *prp4* leads to the aberrant splicing of one or more core clock transcripts. Analysis of clock transcripts indicated an increase in *tim* levels in the late night in flies with reduced *prp4* (*Figure 4—figure supplement 1*), but in order to apply an unbiased approach, we performed RNA-Sequencing (RNA-Seq) analysis of fly heads in which *prp4* had been

**Table 1.** Free-running locomotor behavior of flies expressing RNAi against tri-snRNP components

| Genotype | N | rhythmicity* (%) | Period (hours ± SEM) | Power (FFT ± SEM) |
|---|---|---|---|---|
| TUG; Dcr2/+ | 35 | 100% | 23.80 (0.06) | 0.11 (0.01) |
| pdfGal4, Dcr2/+ | 19 | 100% | 24.06 (0.06) | 0.10 (0.01) |
| elavGal4; Dcr2/+ | 30 | 97% | 23.59 (0.33) | 0.06 (0.03) |
| +/prp4$^{RNAi}$(GD) | 16 | 100% | 23.62 (0.06) | 0.09 (0.01) |
| +/prp4$^{RNAi}$(KK) | 16 | 100% | 23.32 (0.05) | 0.13 (0.01) |
| +/prp3$^{RNAi}$(GD) | 10 | 100% | 23.73 (0.06) | 0.10 (0.01) |
| +/prp3$^{RNAi}$(KK) | 15 | 100% | 23.45 (0.06) | 0.12 (0.01) |
| +/prp8$^{RNAi}$(GD) | 15 | 100% | 23.64 (0.07) | 0.11 (0.01) |
| +/prp31$^{RNAi}$(KK) | 13 | 100% | 23.35 (0.06) | 0.12 (0.04) |
| +/brr2$^{RNAi}$(KK) | 16 | 100% | 23.47 (0.06) | 0.11 (0.01) |
| TUG; Dcr2 > prp4$^{RNAi}$(GD) | 34 | 71%† | 26.55 (0.29)‡ | 0.09 (0.01) |
| TUG; Dcr2 > prp4$^{RNAi}$(KK) | 40 | 45%† | 29.47 (0.48)‡ | 0.09 (0.01) |
| TUG; Dcr2 > prp3$^{RNAi}$(GD) | 11 | 100% | 27.45 (0.49)‡ | 0.08 (0.02) |
| TUG; Dcr2 > prp3$^{RNAi}$(KK) | 33 | 0%† | - | - |
| TUG; Dcr2 > prp8$^{RNAi}$(GD) | 30 | 23%† | 28.00 (1.02)‡ | 0.02 (0.01)‡ |
| TUG;Dcr2 > prp31$^{RNAi}$(KK) | 17 | 88% | 25.33 (0.19)‡ | 0.06 (0.01)¶ |
| TUG;Dcr2 > brr2$^{RNAi}$(KK) | 24 | 0%† | - | - |
| pdfGal4, Dcr2 > prp4$^{RNAi}$(GD) | 30 | 90% | 25.85 (0.32)‡ | 0.06 (0.01)‖ |
| pdfGal4, Dcr2 > prp4$^{RNAi}$(KK) | 38 | 97% | 24.81 (0.32)‡ | 0.09 (0.02) |
| elavGal4; Dcr2 > prp4$^{RNAi}$(GD) | 27 | 52%† | 24.54 (0.14)§ | 0.03 (0.02)# |

* Flies with FFT value >0.01 are considered to be rhythmic.

†p < 0.001 compared to both of the heterozygous controls, by $\chi^2$ analysis.

‡p < 0.001 compared to both of the heterozygous controls, by Student's t test.

§p < 0.001 compared to RNAi control but not significant (p > 0.05) compared to elavGal4; Dcr2/+ control, by Student's t test.

¶p < 0.01 compared to TUG; Dcr2/+ control but not significant (p > 0.05) compared to RNAi control, by Student's t test.

‖p < 0.01 compared to pdfGal4; Dcr2/+ control and p < 0.05 compared to RNAi control, by Student's t test.

#p < 0.05 compared to RNAi control but not significant (p > 0.05) compared to elavGal4; Dcr2/+ control, by Student's t test.

Note: **Table 1** is related to **Figure 3**.

DOI: https://doi.org/10.7554/eLife.39821.006

knocked down. As clock protein expression in the eye contributes a majority of the signal in head assays, we used an eye-specific *Glass Multiple Promoter* (*GMR-Gal4*) driver for *prp4* knockdown. Overall gene expression was dramatically influenced by *prp4* downregulation (433 down, 310 up at FDR < 0.05) (*Supplementary file 2*). Pathway enrichment analysis using DAVID identified changes in folate biosynthesis as well as in other broad pathway categories such as protein export, protein processing in endoplasmic reticulum and drug metabolism (*Supplementary file 1*). Interestingly, components of folate metabolism have been previously implicated in circadian clock regulation in human cells (*Zhang et al., 2009*). Despite the fact that PRP4 is a component of the core spliceosome required for constitutive splicing, we did not detect dramatic effects on global splicing. Using the Comprehensive AS Hunting (CASH) method, which assays for splicing events, our analysis identified 45 genes exhibiting differential splicing upon *prp4* downregulation, with FDR ≤ 0.05 (Wu, W., et al., 2017) (*Supplementary file 3*).

An intron retention event in *tim* that was significantly upregulated upon *prp4* knockdown was of particular interest due to its potential clock function (*Figure 4A–B*). Our initial splicing analysis was performed with CASH, but we additionally ran the Cufflinks-2.2 pipeline to obtain psi (percent

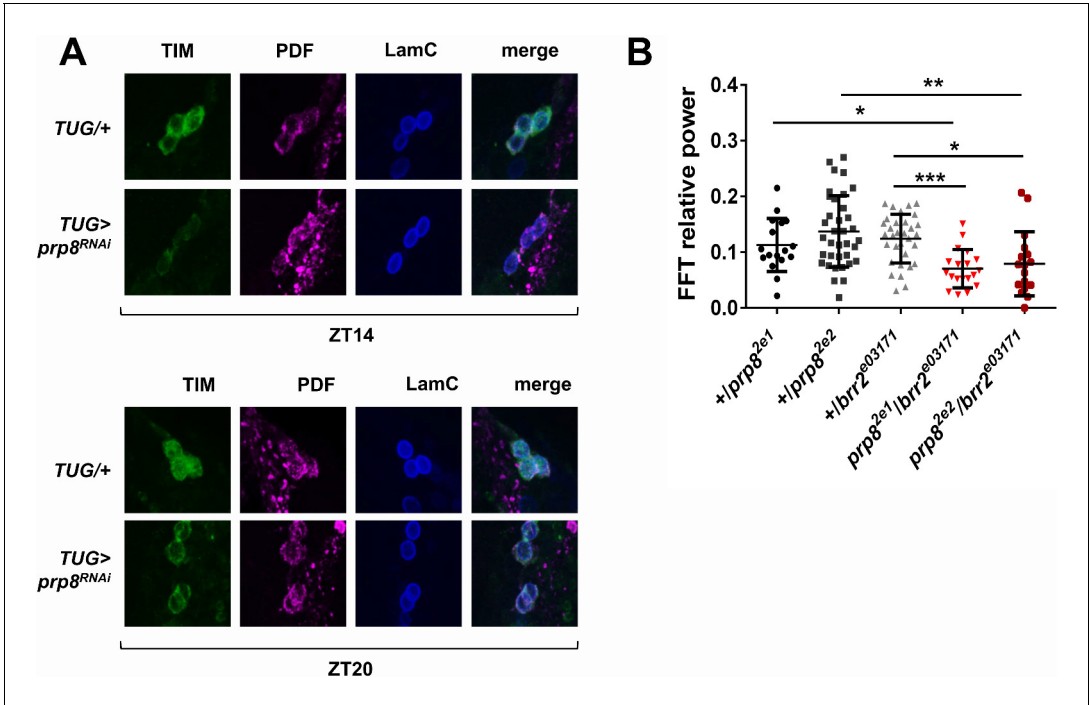

**Figure 3.** PRP8 regulates TIM levels and the strength of rest:activity rhythms. (**A**) TIM levels are decreased in s-LNvs of *prp8* knockdown flies. Adult brains were dissected at ZT14 and ZT20 on the 4th day in LD cycle and immunostained with TIM (green), PDF (magenta) and LaminC (blue) antibodies. Genotypes are indicated on the sides of each panel. Displayed images are representative of two independent experiments. (**C**) Trans-heterozygous *prp8/brr2* mutants have weaker circadian rhythms compared to their heterozygous controls, p* ≤ 0.05, p** ≤ 0.01, p*** ≤ 0.001 by one-way ANOVA, Tukey post hoc test. All FFT value were used in the analysis, including the arrhythmic ones (FFT < 0.01). Error bars represent mean ±SEM (n = 17–36). *Figure 3* is related to *Table 1*.

DOI: https://doi.org/10.7554/eLife.39821.007

spliced in) information for differentially spliced isoforms (*Supplementary file 4*). Importantly, the same retention event in *tim*, at position chr2L:3499764–3500000 (hereafter we refer to this intron as *tim-tiny* for simplicity), was consistently identified as significantly upregulated with both analyses (*Figure 4A*). For comparison, differential alternative splicing in only four other genes (*trol*, *pex7*, *Npc1a*, *vsg*) was consistently identified with both of the algorithms (*Figure 4A*). Moreover, the RNA-Seq reads across the *tiny-tiny* junction normalized to its neighboring junction (2L:3500362–3500422), which is spliced out in all of the *tim* isoforms, were significantly increased in the *prp4* knockdown samples (*GMR >prp4^RNAi*) compared to the controls (*GMR/+*), further pointing to the retention of *tim-tiny*. To estimate how common *tim-tiny* retention was in control flies (*GMR/+*), we quantified the ratio of isoforms that contain *tim-tiny* (*tim*-RM and *tim*-RS) to those that do not contain this intron using our Cufflinks-2.2 output (*Supplementary file 4*). This analysis indicated that the isoforms retaining *tim-tiny* were twice as abundant as other isoforms at ZT 8 (when all of the RNA-Seq samples were collected) (*Supplementary file 5*).

Motivated by the RNA-Seq data, we sought to verify the retention of the *tim-tiny* intron upon pan-neuronal *prp4* knockdown. For this purpose, we designed primers that would amplify only the transcript containing the retained intron or only the spliced transcript respectively (*Figure 4—figure supplement 2*). The ratio of retained to spliced signal would then indicate the relative intron retention. Using this approach, increased *tim-tiny* retention was consistently detected upon pan-neuronal *prp4* knockdown (*Figure 4D*). Additionally, by performing a control set of experiments, the signal from the 'retained' primer set targeting *tim-tiny* was verified not to reflect genomic DNA contamination (*Figure 4—figure supplement 2A–B*). First, retained intron levels were normalized to total *tim* mRNA. As total *tim* mRNA was amplified with primers that do not span junctions, but rather bind within sequences of a different exon ('exon'), they are also expected to detect any contaminating DNA in addition to mRNA. As with the *tim-tiny* retained/spliced ratio, the retained/exon ratio

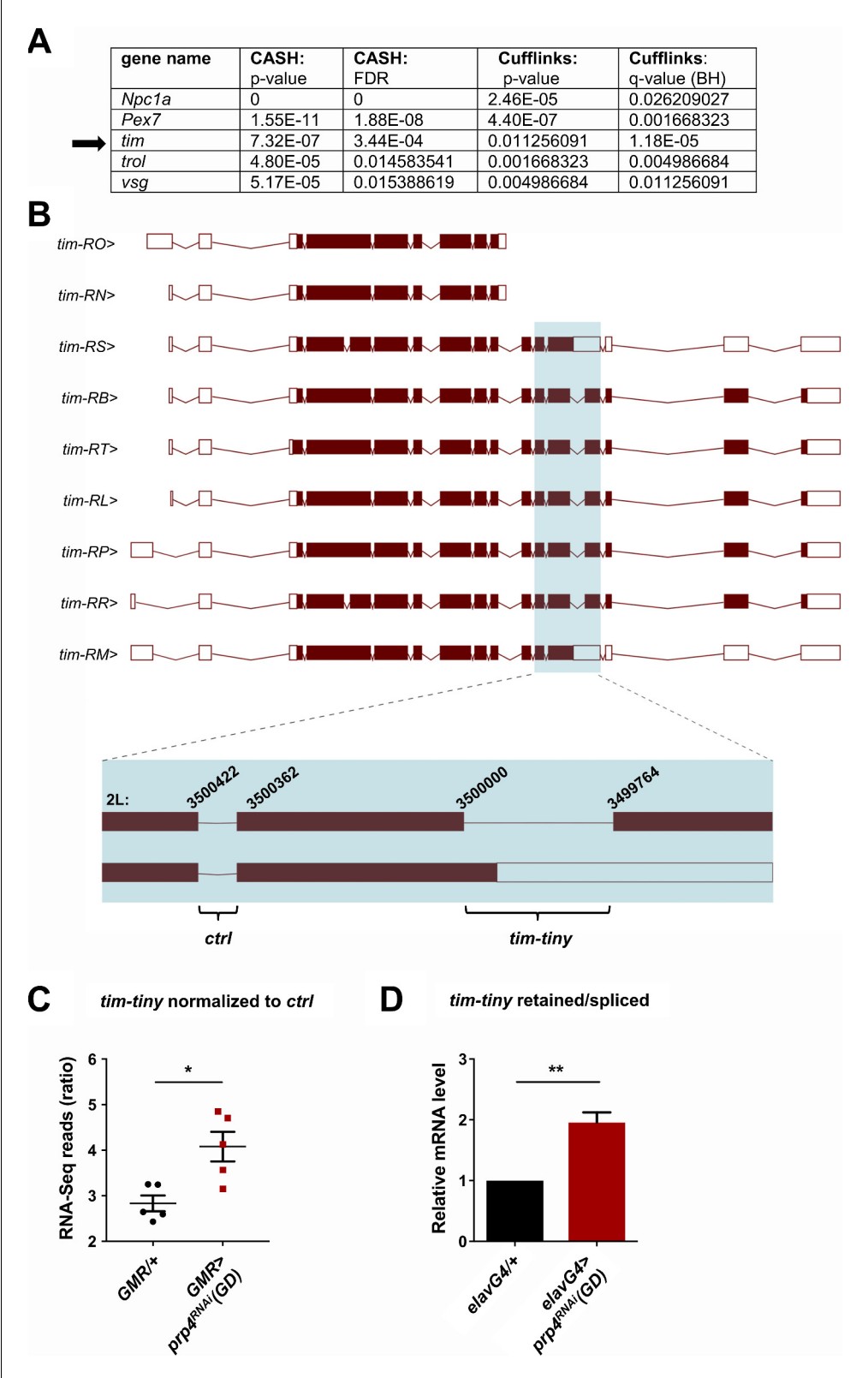

**Figure 4.** PRP4 regulates *tim* splicing. (**A**) Only five genes were identified as differentially spliced upon *prp4* downregulation with both CASH and Cufflinks/differential psi (percent spliced in) pipelines. For each gene, the corresponding p-value and False Discovery Rate (FDR) or q-value as determined by Benjamini-Hochberg (BH) procedure are reported. (**B**) *tim* isoforms (image adapted from Ensembl Fruitfly release 92, genome assembly BDGP6) are displayed. The boxes indicate exons, with filled boxes (brown) representing protein-coding sequences. The region of interest is enlarged

*Figure 4 continued on next page*

*Figure 4 continued*

(blue box) and depicts a constitutively spliced intron ('*ctrl*') and the intron that gets retained upon *prp4* knockdown ('*tim-tiny*'). The chromosomal coordinates of these introns are indicated at their respective exon-intron junctions. (C) *tim-tiny* retention was revealed by RNA-Seq analysis in samples with *prp4* downregulated (*GMR > prp4* RNAi). The number of RNA-Seq reads across the *tim-tiny* intron normalized to the number of reads across the *ctrl* intron is higher in *prp4* knockdown flies (*GMR > prp4^RNAi*) compared to controls (*GMR/+*). Data represent five independent biological replicates. Error bars represent mean RNAiSEM. *p ≤ 0.0001 as determined by CASH (refer to panel A). (D) An increase in intron retention in flies with pan-neuronal *prp4* knockdown was confirmed with qPCR analysis. *Dicer2 (Dcr2)* was co-expressed with the *prp4* RNAi transgene to increase its knockdown efficiency. Data represent four independent biological replicates, with technical triplicates performed during the qPCR step for each replicate. **p ≤ 0.01 to control (*elavGal4; Dcr2/+*) as determined by Student's t test. Data represent mean ±SEM.

DOI: https://doi.org/10.7554/eLife.39821.008

The following figure supplements are available for figure 4:

**Figure supplement 1.** Pan-neuronal knockdown of *prp4* increases *tim* expression.

DOI: https://doi.org/10.7554/eLife.39821.009

**Figure supplement 2.** Analysis of the splicing of *tim-tiny*, *tim-cold* and *per* introns.

DOI: https://doi.org/10.7554/eLife.39821.010

indicated increased *tim-tiny* retention upon *prp4* knockdown, in particular at ZT12, a time point used for our initial RNA-Seq analysis (***Figure 4—figure supplement 2B***). Additionally, we amplified RNA using primers that span an exon-exon junction at a different part of *tim* ('mRNA'), which hence should only detect mRNA, and found that the ratio of *tim* 'exon' to 'mRNA' was not different between the control (*elavGal4; Dcr2/+*) and the *prp4* knockdown flies (*elavGal4; Dcr2 > prp4^RNAi(GD)*), indicating that residual DNA contamination does not contribute to the increased detection of *tim-tiny* with *prp4* knockdown (***Figure 4—figure supplement 2B***).

## Intron retention in *tim* decreases TIM levels and affects circadian behavior

To assay the influence of *prp4*-dependent *tim-tiny* intron retention on TIM levels, we generated *tim* cDNA constructs that lacked the intron ('*tim*-spliced'), included the intron ('*tim*-retained') or included the intron along with a silent T > A mutation in the 5' donor splice site at 2L:3499999 ('*tim*-retained-ssM') (***Figure 5A***). All constructs were transfected into S2 cells and assayed for their ability to drive expression of TIM protein. Intron inclusion had a drastic effect on the total levels of TIM, ranging from highest in the condition when no intron was present to no detectable full size TIM in the condition when splicing was blocked with the mutation (***Figure 5B***). Expression of the construct with the 5' splice site mutation led to low level production of a shorter TIM isoform that we called TIM^tiny. This isoform was entirely absent when *tim* cDNA lacking the *tim-tiny* intron was expressed but was produced upon expression of the cDNA construct that included *tim-tiny*. Therefore, TIM^tiny is a truncated TIM isoform generated from the *tim* mRNA carrying an unspliced *tim-tiny* intron. Total mRNA levels were not different between these three experimental conditions, pointing to the post-transcriptional regulation of TIM protein abundance (data not shown).

Next, we overexpressed *tim* cDNA transgenes (***Figure 5A***) in flies, using a circadian cell driver (*TUG*). Normally, overexpression of *tim* using the Gal4-UAS system reduces rhythmicity and increases free-running periods, likely due to prolonged expression of the excess protein coupled with little negative feedback to downregulate production (***Yang and Sehgal, 2001***). Overexpression of '*tim*-spliced' and '*tim*-retained' cDNAs caused lengthening of free-running periods (***Figure 5D***). However, the circadian behavior of flies overexpressing the cDNA construct with the 5' splice site mutation in *tim-tiny* (*TUG; Dcr2 > tim*-retained + ssM) was not different from that of the controls (*TUG; Dcr2/+*). This would fit with our observations from S2 cells, which suggested a loss of full-sized TIM (TIM^full) expression when *tim-tiny* was primed for selective retention. Thus, we hypothesized that overexpression of the construct with the 5' splice site mutation did not produce TIM^full in flies. We assayed TIM levels in flies overexpressing different *tim* cDNA constructs through western blots of fly head lysates collected at ZT10 when the endogenous TIM levels are low in controls (*TUG; Dcr2/+*) (***Figure 5C***). As predicted, flies that overexpressed '*tim*-retained-ssM' cDNA did not have increased TIM^full relative to controls, although both '*tim*-spliced' and '*tim*-retained' transgenes expressed abundant amounts of protein. Interestingly, we detected a band, likely corresponding to TIM^tiny, in *TUG; Dcr2 > tim*-retained + ssM flies, which was absent in the *TUG; Dcr2/+* controls. These data

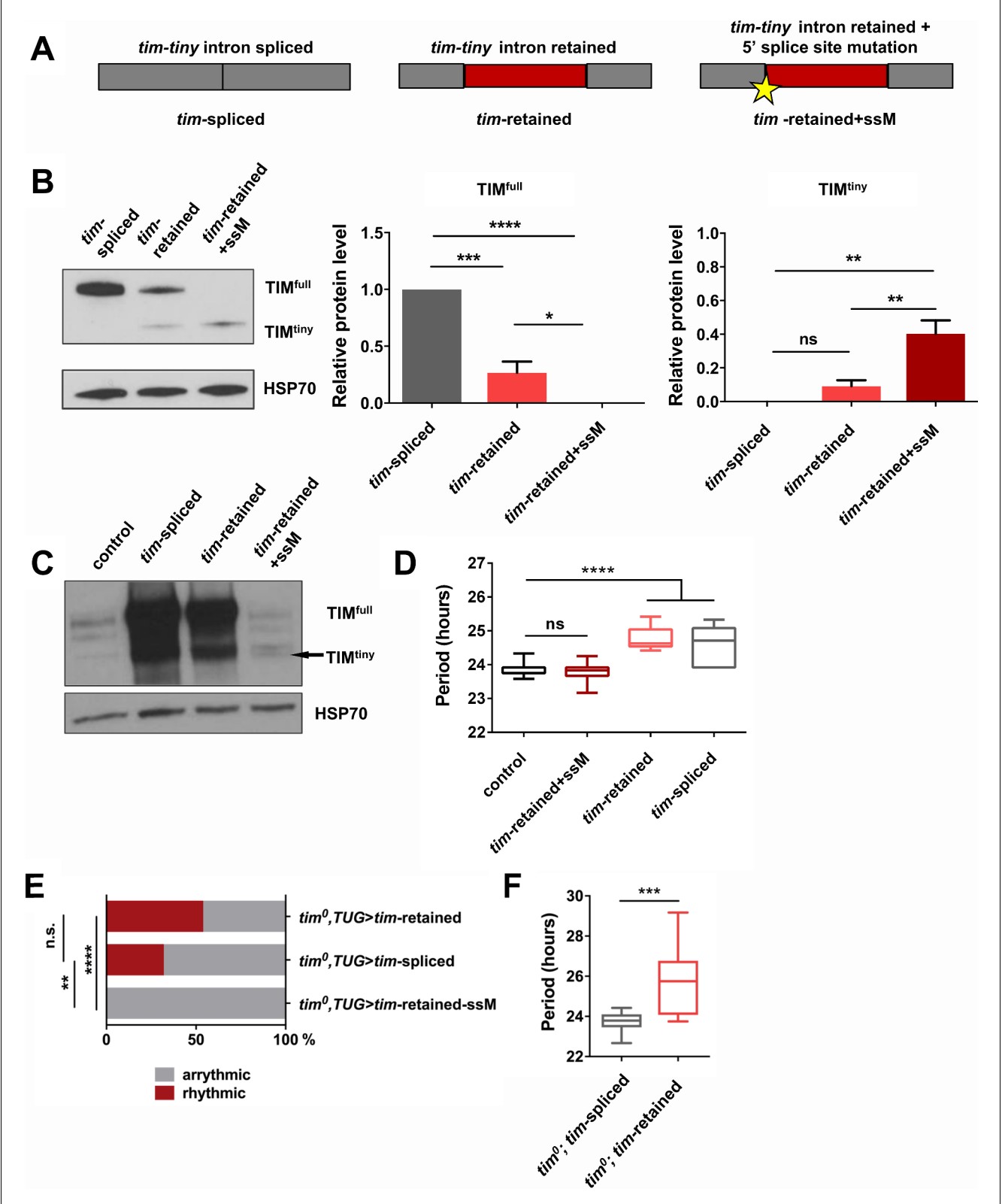

**Figure 5.** Intron retention in *tim* decreases TIM levels and affects circadian behavior. (**A**) Schematic depiction of three *tim* cDNA constructs used to assess the effect of *tim-tiny* retention (red block) on TIM levels. (**B**) Retention of *tim-tiny* intron decreases full-length TIM and leads to production of a minor TIM^tiny isoform. S2 cells were transfected with constructs described in (**A**) and western blots of cell lysates were probed with TIM antibody. Western blots are representative of 3 independent experiments. In the panels on the left, total levels of TIM isoforms upon expression of splice-specific

*Figure 5 continued on next page*

*Figure 5 continued*

cDNA constructs were quantified. TIM levels were normalized to HSP70 and expressed relative to the TIM[full] levels in cells overexpressing a fully spliced (*tim*-spliced) construct. *p ≤ 0.05, **p ≤ 0.01, ***p ≤ 0.001, ****p ≤ 0.0001 by one-way ANOVA and Holm-Sidak's post hoc test. Data represent mean ±SEM (n = 3). (C) Western blots of head lysates of flies overexpressing *tim* cDNA with a 5' splice site mutation in *tim-tiny* (TUG; Dcr2 > *tim*-retained+ssM) reveal production of TIM[tiny] (arrow) and decrease in TIM[full] compared to flies overexpressing intronless *tim* cDNA (TUG; Dcr2 > *tim*spliced) or *tim* cDNA that includes *tim-tiny* ((TUG; Dcr2 > tim retained). All flies were collected at ZT10, when endogenous TIM levels are low in control flies (TUG; Dcr2/+). Western blots are representative of 4 independent experiments. (D) Flies overexpressing *tim* cDNA constructs with 5' splice site mutation (TUG; Dcr2 >*tim*-retained+ssM) do not lengthen circadian period. n = 6–26; n.s., not significant at the 0.05 level; ****p ≤ 0.0001 to control (TUG; Dcr2/+) by one-way ANOVA and Tukey's post hoc test. (E) *TUG*-driven expression of *tim* cDNA, with both '*tim*-spliced' and '*tim*-retained' constructs, rescues circadian rhythms in *tim[0]* flies. n.s., not significant at the 0.05 level, **p ≤ 0.01, ****p ≤ 0.0001 by pairwise Fischer's exact test (n = 28–41). (F) *TUG*-driven rescue of *tim[0]* circadian rhythms with *tim* cDNA lacking *tim-tiny* (*tim[0],TUG > tim* spliced) results in shorter periods than with *tim* cDNA that includes *tim-tiny* (*tim[0],TUG > tim* retained). ***p ≤ 0.001 by Student's t test, n = 10–22.

DOI: https://doi.org/10.7554/eLife.39821.011

further corroborate our S2 cell findings and suggest that selective *tim-tiny* retention acts to reduce overall TIM levels.

To further understand the effect of *tim-tiny* splicing on circadian behavior, we overexpressed '*tim*-spliced', '*tim*-retained' and '*tim*-retained-ssM' using the *TUG* driver in the *tim[0]* homozygous background. We hypothesized that differential splicing of *tim-tiny* is necessary for the maintenance of circadian rhythmicity and so the '*tim*-retained' construct would rescue the behavioral rhythm most efficiently because it allows for both the splicing and retention of the intron. Additionally, because *prp4* knockdown increases the retention of *tim-tiny* (*Figure 4*) and prolongs the rhythm (*Figure 1*), we speculated that the '*tim*-spliced' construct would rescue with a shorter period than the '*tim*-retained' construct. As expected, the expression of *tim* cDNA with a 5' splice site mutation in *tim-tiny* ('*tim*-retained + ssM') did not restore rhythms in *tim[0]* flies (*Figure 5E*). The other two cDNA constructs, '*tim*-retained' and '*tim*-spliced' rescued rhythms in 54% and 32% of *tim[0]* flies, respectively. Importantly, there was a significant difference in period length between the flies that were rhythmic (*Figure 5F*). As discussed above, the UAS-GAL4 system typically over-expresses proteins and so rescues *per/tim* mutants with longer periods (*Yang and Sehgal, 2001*), which we observed for the '*tim*-retained' isoform (~26 hr). Shorter periods were seen with the '*tim*-spliced' version (*Figure 5F*), supporting the idea that *tim-tiny* retention promotes clock delays.

## *tim* splicing is regulated by the clock and by temperature

We next asked if the splicing of *tim* is under circadian regulation. For this purpose, we generated relative intron retention profiles across regular intervals under both LD and DD conditions (*Figure 6A–B*). In LD conditions, intron retention of *tim-tiny* did not display a significant cycle (by JTK analysis) but nevertheless showed a peak at ZT8. In DD conditions, interestingly, we observed robust cycling of *tim-tiny* with a crest at CT8. It was previously reported that the last (~850 bp) intron in *tim* (also known as *tim-cold*) is also sometimes retained (*Boothroyd et al., 2007*). To determine if the retention of *tim-cold* is rhythmic, as is the retention of *tim-tiny*, we profiled *tim-cold* intron retention across a 24 hr cycle under different conditions. We found that *tim-cold* retention cycled in both LD and DD with similar phases, such that peak intron retention was during the day in LD and the subjective day in DD. To further verify that the cycling we detected for *tim* splicing was under clock control, we assayed *tim-tiny* and *tim-cold* intron retention profiles in *per[01]* mutant flies. In *per[01]* flies under both LD and DD conditions, the cycling of both *tim-tiny* and *tim-cold* intron retention was abolished, indicating that the circadian clock drives circadian oscillations in *tim* splicing. We suggest that retention of the *tim-tiny* intron during the day serves to delay the accumulation of TIM protein, in particular when light is not available to degrade TIM.

It was previously reported that *tim-cold* retention is regulated by temperature and peaks at colder temperatures (*Boothroyd et al., 2007*). To determine if the splicing of *tim-tiny* was similarly sensitive to temperature, we used a temperature entrainment paradigm (12 hr:12 hr 30°C:25°C) under constant photic conditions (*Figure 6C–D*). In constant dark, temperature cycles were able to drive *tim* expression as previously reported (Glaser, F.T., and Stanewsky, R., 2005). Additionally, the profile of *tim-cold* retention was cyclic with the highest retention levels during the colder (25°C) temperatures. Interestingly, *tim-tiny* retention was robustly rhythmic under these temperature

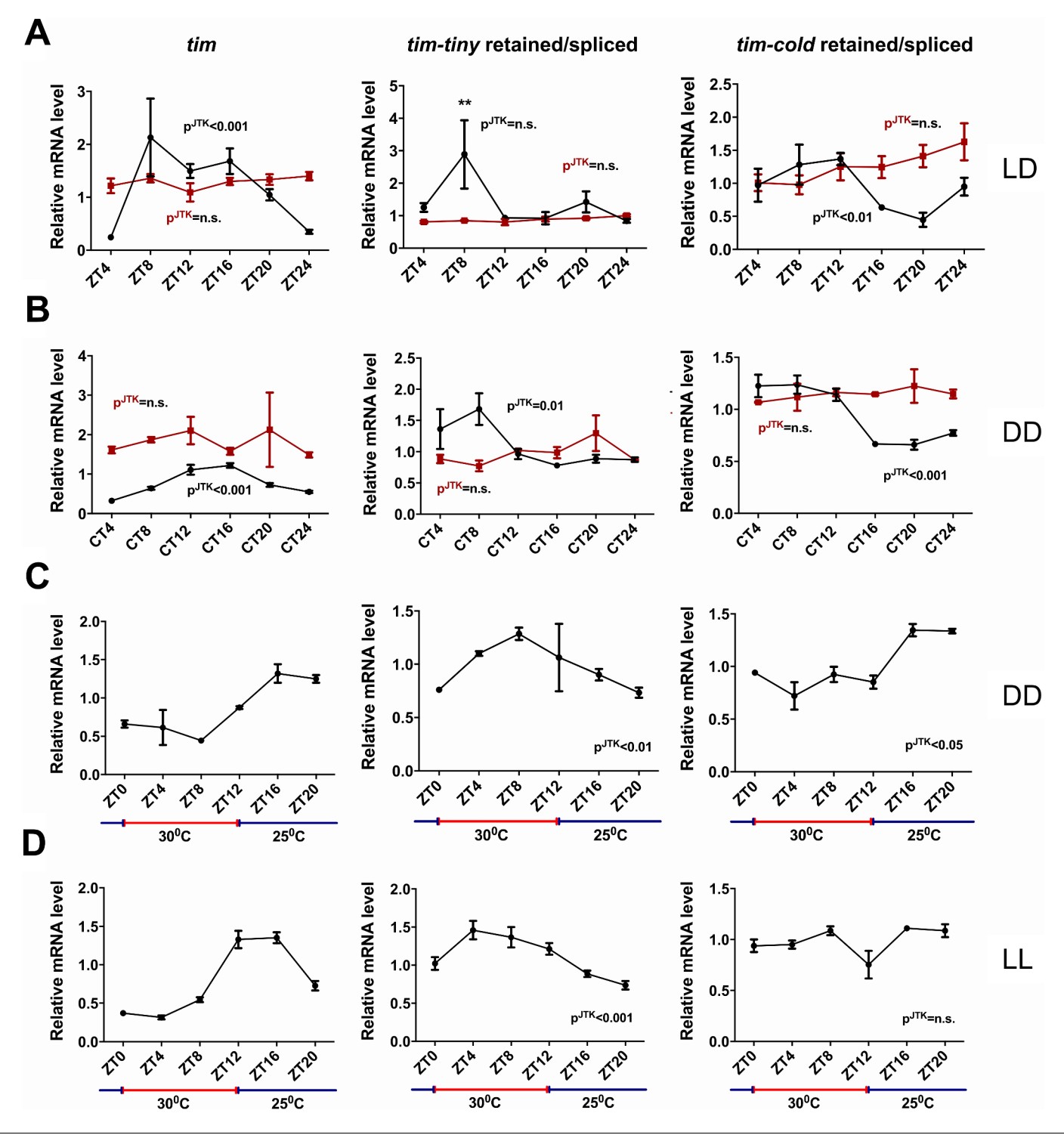

**Figure 6.** *tim* splicing is regulated by the clock and by temperature. Flies were entrained for at least 3 days in 12 hr:12 hr light:dark (LD) conditions and collected in LD (A) or on the first day of transfer to constant darkness (B) at indicated ZT or CT time points, respectively. Splicing of *tim-tiny* and *tim-cold* introns was quantified as a ratio of retained to spliced levels using qPCR analysis. Three independent qPCR experiments were performed in triplicate, normalized to *rp49* and analyzed using the ΔΔCt method. $p^{JTK}$ indicates cycling as assessed by JTK cycle analysis for wild-type *iso[31]* flies (black) and *per[01]* circadian mutants (red). *tim-tiny* intron retention is increased in LD at ZT8 (A) as calculated by two-way ANOVA and Sidak's post hoc test, p** ≤ 0.01. Wild-type *iso[31]* flies were collected at indicated ZT time points after at least four full days of entrainment in 12 hr:12 hr 30C:25C temperature cycles in constant dark (C) or constant light (D) conditions. Splicing of *tim-tiny* and *tim-cold* introns was quantified as a ratio of retained to

*Figure 6 continued on next page*

*Figure 6 continued*

spliced levels using qPCR analysis. Three independent qPCR experiments were performed in triplicate, normalized to *rp49* and analyzed using the ΔΔCt method. p^JTK indicates cycling as assayed by JTK cycle analysis. Error bars = SEM.

DOI: https://doi.org/10.7554/eLife.39821.012

conditions, yet, unlike *tim-cold*, intron retention was increased by higher temperatures and decreased by lower ones. Thus, higher temperatures, which are typically associated with daytime hours, may reduce TIM levels by regulating the splicing of *tim-tiny*, while light does so through TIM degradation (*Hunter-Ensor et al., 1996*; *Myers et al., 1996*; *Zeng et al., 1996*; *Naidoo et al., 1999*). We observed the same relationship and even more robust cycling of *tim-tiny* under temperature cycles (12 hr:12 hr 30°C:25°C) in constant light conditions (*Figure 6D*). In these conditions, *tim-cold* did not cycle, further indicating different regulation of these two splicing events in *tim*.

## Discussion

In this study, we identify a novel alternative splicing (AS) mechanism that affects the pace of endogenous circadian oscillations and implicates PRP4 and other tri-snRNP components in circadian clock regulation. Importantly, our findings contribute to the understanding of a longstanding question in the circadian field, specifically how negative feedback by clock proteins is delayed in order to permit distinct phases, and therefore oscillations, of transcriptional activation and repression. In Drosophila, PER translation does not appear to be delayed relative to its mRNA production in a daily cycle (*Chen et al., 1998*), but the protein is initially destabilized through its phosphorylation by the DOUBLETIME (DBT) kinase (*Price et al., 1998*). TIM stabilizes PER by alleviating this effect of DBT, and so accumulation of TIM, which only occurs after dark as TIM is degraded by light, determines the rise of PER. How this mechanism persists when light is not a cycling cue, for instance in constant darkness or in temperature cycles in constant light, was not known. Alternative splicing of the *tim-tiny* intron may be critical under these conditions.

The circadian profile of *tim-tiny* intron retention (i.e. high during the daytime) is consistent with it delaying the accumulation of TIM protein. Indeed, this mechanism is particularly robust in DD (*Figure 6*) and also in temperature cycles in constant light conditions (*Glaser and Stanewsky, 2005*; *Yoshii et al., 2005*). The retention of *tim-tiny* peaks at high temperatures, which correlates with low TIM levels (Glaser, F.T., and Stanewsky, R., 2005; *Yoshii et al., 2005*). We propose a model (*Figure 7*) whereby the splice choice at the *tim-tiny* locus modulates the rate of TIM accumulation, which in turn affects the total levels of nuclear PER and TIM. The intron retention of *tim-tiny*, therefore, likely contributes to a delay between initial *tim* expression and accumulation of TIM, ensuring stability and robustness of the circadian oscillator. At the same time, the response of this splicing event to temperature promotes flexibility of the clock.

Traditionally, it has been accepted that AS is driven by a set of auxiliary splicing factors, such as serine/arginine-rich (SR) proteins and heterogeneous nuclear ribonucleoproteins (hnRNPs), that act on *cis*-regulatory splicing modules and recruit U1 and U2 snRNP machinery (*Matera and Wang, 2014*; *Bradley et al., 2015*; *Han et al., 2011*). However, studies from a range of organisms increasingly point to the involvement of the core spliceosomal machinery not only in AS execution but also in splice junction selection (*Brooks et al., 2015*; *Burckin et al., 2005*; *Clark et al., 2002*; *Park et al., 2004*; *Pleiss et al., 2007*). Our RNA-seq findings further support the idea that the abundance/stability of at least some components of the core spliceosome has an effect on a select subset of AS events, while constitutive splicing seems to be unperturbed (*Supplementary file 3* and *4*). A functional network of spliceosomal proteins has been proposed on the basis of their knockdown phenotypes (*Papasaikas et al., 2015*). According to this model, PRP4 resides within the tri-snRNP regulatory module, which largely coincides with the previously reported physical interactions between the components of the tri-snRNP (*Dellaire et al., 2002*; *Bottner et al., 2005*; *Schneider et al., 2010*). Our findings of similar circadian behavioral effects caused by loss of any of several tri-snRNP components strongly implicate the tri-snRNP complex in circadian regulation (*Table 1*). It remains to be established how the decreased abundance of tri-snRNP components triggers changes in alternative splicing. We hypothesize that tri-snRNP level/activity is limiting for a subset of AS reactions, likely the ones with weaker splice sites. This hypothesis is supported by two

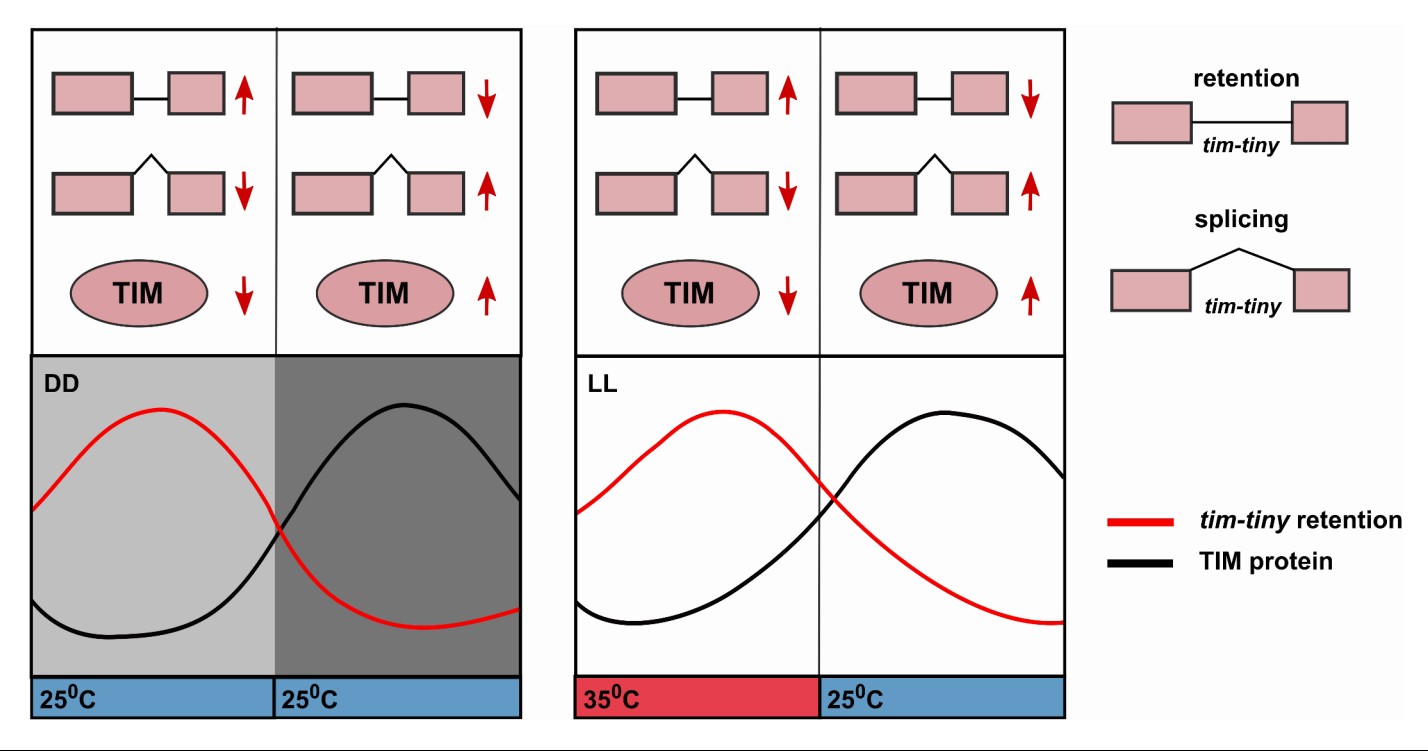

**Figure 7.** Alternative splicing of the *tim*-tiny intron promotes oscillations of TIM levels. The model depicts how retention of the *tim-tiny* intron, which is increased upon downregulation of *prp4*, regulates TIM cycling. Both the circadian clock (**A**) and temperature cycles (**B**) regulate retention of the *tim-tiny* intron. (**A**) Increased retention of *tim-tiny* during the subjective day (light gray) in dark:dark (DD) conditions serves to decrease TIM levels and delay the accumulation of TIM in the absence of light. (**B**) Higher temperatures, typically associated with daytime hours, increase *tim-tiny* retention. Temperature cycles can maintain clock function under constant light conditions (LL), which would otherwise disrupt the clock. Entrainment by temperature appears to be driven by a reduction of TIM protein at the higher temperature (Yoshii, T., et al., 2005). We propose that under temperature cycles, retention of *tim-tiny* sets the levels of TIM and contributes to maintenance of the molecular clock.

DOI: https://doi.org/10.7554/eLife.39821.013

recent studies reporting that (1) *Prp4* in fission yeast is necessary to recognize and splice the introns with weak splice sites (*Eckert et al., 2016*) and (2) decreased availability of mammalian *Prpf8* leads to the selective retention of introns that harbor weak 5' splice sites (Wickramasinghe, V.O., et al., 2015).

We speculate that tri-snRNP components constitute a well-conserved regulatory module for circadian clocks. Conservation of the circadian role of tri-snRNP is suggested by several findings. First, the human homolog of PRP4 was identified as a hit in a genome-wide RNAi screen for regulators of the circadian clock (*Zhang et al., 2009*). Secondly, SM-like (LSM) proteins that are associated with U6 snRNP were recently shown to regulate circadian rhythmicity in both *Arabidopsis* and in mammalian cell culture (*Perez-Sántángelo et al., 2014*). Finally, some tri-snRNP components (Prpf8, Prpf31 and SART1) physically associate with mammalian PER2 complexes, further highlighting potential cross-talk between central clock components and the tri-snRNP (*Kim et al., 2014*).

Although AS has been implicated in the regulation of circadian clocks, it has not been linked to clock function in the manner we report here. In *Neurospora crassa*, the ratio of alternatively spliced *frequency (frq)* isoforms determines the robustness of circadian rhythmicity and fine-tunes the period length (*Garceau et al., 1997*; *Liu et al., 1997*). In *Arabidopsis thaliana*, AS regulates the circadian clock by multiple mechanisms, such as the production of new isoforms that competitively inhibit functional clock proteins (*Seo et al., 2012*) and the modulation of clock RNA levels via the nonsense-mediated decay (NMD) pathway (*James et al., 2012*; *Kwon et al., 2014*). In *Drosophila*, the alternative splicing regulator SR-related matrix protein of 160 kDa (SRm160) modulates PER levels locally in the pacemaker neurons to regulate circadian rhythms (*Beckwith et al., 2017*). The key theme that emerges from these studies is that AS acts directly on core clock components to set their

levels. Alternatively, splicing mechanisms could regulate diurnal rhythmicity of neuronal excitability, as proposed for splicing of BK channels in the suprachiasmatic nucleus (*Shelley et al., 2013*).

Our study identifies a novel splicing event in *tim* that can regulate TIM levels in both cell culture and in flies (*Figure 5*). How does the splicing affect TIM levels? It is unlikely that the retention of *tim-tiny* leads to NMD-mediated RNA decrease because the flies with *prp4* downregulated do not have reduced *tim* levels (*Figure 4—figure supplement 1*). On the contrary, overall *tim* RNA levels tend to be increased in those flies. Additionally, expression of *tim* cDNA constructs with constitutively retained *tim-tiny* (*Figure 5*) decreases TIM levels without altering *tim* mRNA levels (data not shown). Therefore, we suggest that it is either the *tim* mRNA translation step or the stability of the truncated TIM isoform, TIM^tiny, produced by *tim-tiny* that is sub-optimal. The 267 amino acids at the C-terminus of TIM that are predicted to be lost in TIM^tiny include a putative cytoplasmic localization signal (*Saez et al., 2011*) as well as a predicted threonine phosphorylation site (*Bodenmiller et al., 2007*), both of which could significantly change stability and function of TIM. Notably, TIM^tiny is typically not detected in western blots of TIM expression in flies, so it appears that *tim-tiny* retention serves only to reduce the amount of *tim* RNA that can effectively produce protein.

While studying the effect of *tim-tiny* retention in isolation can give us a quick snapshot of its importance, overall splicing of *tim* is considerably more complex. In parallel to *tim-tiny*, in this study we examined the splicing profile of a previously reported *tim-cold* intron (*Boothroyd et al., 2007*). Cycles of *tim-tiny* and *tim-cold* intron retention are similar under light:dark and constant dark conditions (*Figure 6*), but temperature cycles have different effects on the splicing of these two introns. *tim-tiny* intron retention increases with hot temperature, whereas *tim-cold* intron retention increases at the onset of the cold cycle (*Figure 6*). Also, *tim-cold* retention does not cycle in temperature cycles in LL, suggesting that it does not contribute to rhythmicity under these conditions. *tim-tiny* intron is upstream of *tim-cold*, which means that *tim-tiny* retention should lead to TIM downregulation regardless of the splicing decision at the *tim-cold* locus. Following differential splicing at *tim-tiny* locus, *tim-cold* could get either retained or spliced, introducing an additional regulatory layer. The interplay of AS at the level of these two introns can produce a range of TIM isoforms, the roles of which remain to be elucidated, particularly with respect to temperature entrainment.

In Drosophila, splicing of *D. melanogaster per* intron 8 (*dmip8*), the intron located in the 3′ untranslated region (UTR) of *per*, is regulated by both light and temperature (*Collins et al., 2004*; *Majercak et al., 2004*; *Majercak et al., 1999*). This splicing mechanism allows flies to delay their evening behavior during long photoperiods and/or high temperatures, and might play a role in seasonal adaptation. On a molecular level, *dmip8* retention delays *per* mRNA and PER protein accumulation. While *dmip8* retention was not identified in our initial RNA-Seq data (*Supplementary file 3* and *4*), our follow up qPCR splicing analysis suggested that PRP4 modestly regulates *per* splicing (*Figure 4—figure supplement 2C*). *dmip8* retention has a small effect on the free-running period length (~25 hr period) (*Cheng et al., 1998*; *Majercak et al., 1999*), but it cannot fully account for the period lengthening phenotype we report for flies with downregulated *prp4* (*Figure 1A–B*; *Table 1*).

Studies in *Arabidopsis* suggest that the expression of a number of tri-snRNP components is regulated by the circadian clock (*Perez-Santángelo et al., 2013*). In Drosophila, recent profiling of mRNA cycling in different neuronal clusters detected *prp4* mRNA oscillations in DN1 clock neurons and *brr2* cycling in LNvs (*Abruzzi et al., 2017*). While these RNA-Seq findings have not been verified through other approaches, they lead us to hypothesize that diurnal oscillations in tri-snRNP components drive circadian splicing of *tim* (*Figure 6*) and potentially other circadian output genes locally in specific clock neurons. This hypothesis is further strengthened by another recent report (*Wang et al., 2018*) that suggests heterogeneous alternative splicing profiles for different circadian neuronal groups. In addition to changes in total levels of PRP4, circadian regulation of its kinase activity could contribute to differential splicing of *tim* over the course of the day. We establish a role for PRP4 in LNvs, the key pacemaker cluster necessary for the maintenance of circadian cycles under constant dark conditions (*Figure 1*). However, based on our findings that splicing of *tim* is regulated by temperature and persists in constant light (*Figure 6*), we speculate that PRP4 also functions in DN1s, clock cells implicated in temperature sensing and entrainment (*Yadlapalli et al., 2018*; *Zhang et al., 2010*). In summary, while much of the focus in the circadian field has been on transcriptional or post-translational control, our findings indicate a critical role for alternative splicing, perhaps in a cell-type-specific manner.

# Materials and methods

## Key resources table

| Reagent type (species) or resource | Designation | Source or reference | Identifiers | Additional information |
|---|---|---|---|---|
| Gene (Drosophila melanogaster) | *prp4* | NA | FLYB:FBgn0027587 | |
| Gene (Drosophila melanogaster) | *timeless (tim)* | NA | FLYB:FBgn0014396 | |
| Gene (Drosophila melanogaster) | *period (per)* | NA | FLYB:FBgn0003068 | |
| Gene (Drosophila melanogaster) | *prp8* | NA | FLYB:FBgn0033688 | |
| Gene (Drosophila melanogaster) | *brr2* | NA | FLYB:FBgn0263599 | also known as *l(3)72Ab* |
| Gene (Drosophila melanogaster) | *prp3* | NA | FLYB:FBgn0036915 | |
| Gene (Drosophila melanogaster) | *prp31* | NA | FLYB:FBgn0036487 | |
| Strain, strain background (Drosophila melanogaster) | $iso^{31}$ | from laboratory stocks | NA | |
| Genetic reagent (Drosophila melanogaster) | $per^{01}$ | Bloomington Drosophila Stock Center (BDSC) | FLYB:FBal0013649 | |
| Genetic reagent (Drosophila melanogaster) | $tim^{0}$ | BDSC | FLYB:FBal0035778 | |
| Genetic reagent (Drosophila melanogaster) | *TUG (Tim-UAS-Gal4)* | BDSC | FLYB:FBtp0011839 | |
| Genetic reagent (Drosophila melanogaster) | *pdfGal4; pdfG4* | BDSC | FLYB:FBtp0011844 | |
| Genetic reagent (Drosophila melanogaster) | *elavGal4; elavG4* | BDSC | BDSC:25750 | |
| Genetic reagent (Drosophila melanogaster) | *GMRGal4; GMR* | BDSC | FLYB:FBti0002994 | |
| Genetic reagent (Drosophila melanogaster) | $prp4^{RNAi}(GD)$ | Vienna Drosophila Resource Center (VDRC) | VDRC:27808 | |
| Genetic reagent (Drosophila melanogaster) | $prp4^{RNAi}(KK)$ | VDRC | VDRC:107042 | |
| Genetic reagent (Drosophila melanogaster) | $prp8^{RNAi}(GD)$ | VDRC | VDRC:18565 | |

*Continued on next page*

*Continued*

| Reagent type (species) or resource | Designation | Source or reference | Identifiers | Additional information |
|---|---|---|---|---|
| Genetic reagent (Drosophila melanogaster) | *prp3*<sup>*RNAi*</sup>*(GD)* | VDRC | VDRC:25547 | |
| Genetic reagent (Drosophila melanogaster) | *prp3*<sup>*RNAi*</sup>*(KK)* | VDRC | VDRC:103628 | |
| Genetic reagent (Drosophila melanogaster) | *prp31*<sup>*RNAi*</sup>*(KK)* | VDRC | VDRC:103721 | |
| Genetic reagent (Drosophila melanogaster) | *brr2*<sup>*RNAi*</sup>*(KK)* | VDRC | VDRC:110666 | |
| Genetic reagent (Drosophila melanogaster) | *prp8*<sup>*2e1*</sup> | BDSC | FLYB: FBal0190235; BDSC:25905 | |
| Genetic reagent (Drosophila melanogaster) | *prp8*<sup>*2e2*</sup> | BDSC | FLYB:FBal0190015; BDSC:25912 | |
| Genetic reagent (Drosophila melanogaster) | *brr2*<sup>*e03171*</sup> | BDSC | FLYB:FBti0041681; BDSC:18127 | |
| Genetic reagent (Drosophila melanogaster) | *UAS-Dicer2; Dcr2* | BDSC | FLYB:FBtp0036672 | |
| Genetic reagent (Drosophila melanogaster) | UAS-*tim*-spliced; *tim*-spliced | this paper | NA | generated by the site-specific PhiC31 Integration System (Rainbow Transgenics) using the attP on the 3<sup>rd</sup> chromosome; pUAST-*tim*-spliced plasmid was used for injection |
| Genetic reagent (Drosophila melanogaster) | UAS-*tim*-retained; *tim*-retained | this paper | NA | generated by the site-specific PhiC31 Integration System (Rainbow Transgenics) using the attP on the 3<sup>rd</sup> chromosome; pUAST-*tim*-retained plasmid was used for injection |

*Continued on next page*

*Continued*

| Reagent type (species) or resource | Designation | Source or reference | Identifiers | Additional information |
|---|---|---|---|---|
| Genetic reagent (Drosophila melanogaster) | UAS-*tim*-retained+ssM; *tim*-retained+ ssM | this paper | NA | generated by the site-specific PhiC31 Integration System (Rainbow Transgenics) using the attP on the 3rd chromosome; pUAST-*tim*-retained+ssM plasmid was used for injection |
| Cell line (Drosophila melanogaster) | S2 | ATCC (Manassas, VA) | FLYB:FBtc0000181; RRID:CVCL:Z992 | |
| Antibody | guinea pig anti-PER (UP1140) | *Garbe et al., 2013* | NA | 1:1000 |
| Antibody | rat anti-TIM (UPR42) | *Jang et al., 2015* | NA | 1:1000 |
| Antibody | rabbit anti-PDF (HH74) | *Garbe et al., 2013* | NA | 1:500 |
| Antibody | mouse anti-LaminC | Developmental Studies Hybridoma Bank (DSHB) | LC28.26 | 1:500 |
| Antibody | mouse anti-HSP70 | Sigma | Cat# H5147 | 1:5000 |
| Recombinant DNA reagent | pIZ/V5-His plasmid | ThermoFischer | Cat# V800001 | backbone |
| Recombinant DNA reagent | pBluescript-*tim* | lab collection | NA | *tim* sequence contained *tim-tiny*; used for subcloning |
| Recombinant DNA reagent | *tim*-spliced; pIZ-*tim*-spliced | this paper | NA | *tim* cDNA was subcloned into pIZ-V5 plasmids |
| Recombinant DNA reagent | *tim*-retained; pIZ-*tim*-retained | this paper | NA | *tim-tiny* intron was subcloned into pIZ-*tim*-spliced vector from pBluescript-*tim* plasmid |
| Recombinant DNA reagent | *tim*-retained+ ssM; pIZ-*tim*-retained+ssM | this paper | NA | generated by mutagenesis of the 5' splice donor site of *tim-tiny* intron from pIZ-*tim*-retained plasmid |
| Recombinant DNA reagent | *tim*-spliced; pUAST-*tim*-spliced | this paper | NA | *tim* cDNA was subcloned from pIZ-*tim*-spliced into pUAST-attB vector |

*Continued on next page*

*Continued*

| Reagent type (species) or resource | Designation | Source or reference | Identifiers | Additional information |
|---|---|---|---|---|
| Recombinant DNA reagent | *tim*-retained; pUAST-*tim*-retained | this paper | NA | *tim* cDNA was subcloned from pIZ-*tim*-retained into pUAST-attB vector |
| Recombinant DNA reagent | *tim*-retained + ssM; pUAST-*tim*-retained+ssM | this paper | NA | *tim* cDNA was subcloned from pIZ-*tim*-retained+ssM into pUAST-attB vector |
| Sequence-based reagent | *tim* PP11542 ('mRNA') _F | ATGGACTGGT TACTAGCAACTCC | | |
| Sequence-based reagent | *tim* PP11542 ('mRNA') _R | GGTCCTCATA GGTGAGCTTGT | | |
| Sequence-based reagent | *per*_F | CGTCAATCC ATGGTCCCG | | |
| Sequence-based reagent | *per*_R | CCTGAAAGA CGCGATGGTG | | |
| Sequence-based reagent | *clk*_F | GGATGCCAAT GCCTACGAGT | | |
| Sequence-based reagent | *clk*_R | ACCTACGA AAGTAGCCCACG | | |
| Sequence-based reagent | *prp4*_F | CACAAGCA GCATCTTTGTATGG | | |
| Sequence-based reagent | *prp4*_R | TGTGGAGTC CCACATTCTTG | | |
| Sequence-based reagent | *tim-tiny*_retained_F | AAACGTGAG TTAAAGTCAACC | | |
| Sequence-based reagent | *tim-tiny*_retained_R | GAGAGGCAC ACAGCATATC | | |
| Sequence-based reagent | *tim-tiny*_spliced_F | CCGCTGGAC AAACTCAACCTC | | |
| Sequence-based reagent | *tim-tiny*_spliced_R | TCGGTATCGC CGAGATCCACG | | |
| Sequence-based reagent | *tim-cold*_retained_F | GGCTCATGA TCATTGCAGCAGC | | |
| Sequence-based reagent | *tim-cold*_retained_R | ATAGTGGG GCACCCGGATCTC | | |
| Sequence-based reagent | *tim-cold*_spliced_F | TTAAACAGCG ACAATGTCTCTTTGG | | |
| Sequence-based reagent | *tim-cold*_spliced_R | GAATTGGATCC TCAGTGATAGTGGG | | |
| Sequence-based reagent | *tim*_non_spanning ('exon')_F | GAAGAACAACG ATATTGTGGGAAAG | | |
| Sequence-based reagent | *tim*_non_spanning ('exon')_R | AGTGGGAGT TGTCAGCAAAG | | |
| Sequence-based reagent | *per*_retained_F | GAGGACCA GACACAGCACGG | | |
| Sequence-based reagent | *per*_retained_R | CGGAGGCAA TTGCTCACTCGT | | |
| Sequence-based reagent | *per*_spliced_F | GAGGACCA GACACAGCACGG | | |
| Sequence-based reagent | *per*_spliced_R | TCGCGTTGA TTCGAAGAATCGTT | | |

*Continued on next page*

*Continued*

| Reagent type (species) or resource | Designation | Source or reference | Identifiers | Additional information |
|---|---|---|---|---|
| Sequence-based reagent | *rp49*_F | GACGCTTCAAG GGACAGTATCTG | | |
| Sequence-based reagent | *rp49*_R | AAACGCGGT TCTGCATGAG | | |
| Sequence-based reagent | *tim_tiny*SSdonorT > A_F | CTGGACAAACG AGAGTTA AAGTCAACC | | |
| Sequence-based reagent | *tim_tiny*SSdonor T > A_R | CGGTCCCA GCTTTTTGGC | | |
| Commercial assay or kit | RNeasy Plus Mini Kit | Qiagen | Cat# 74134 | |
| Commercial assay or kit | Superscript II Reverse Transcriptase | ThermoFischer | Cat# 18064014 | |
| Commercial assay or kit | TRIzol Reagent | ThermoFischer | Cat# 15596026 | |
| Commercial assay or kit | Q5 Site-Directed Mutagenesis Kit | NEB | Cat# E0554S | |
| Commercial assay or kit | Effectene Transfection Reagent | Qiagen | Cat# 301425 | |
| Software, algorithm | Graphpad Prism v7 | Graphpad Software | https://www.graphpad.com/ | |
| Software, algorithm | JTK_CYCLE v3 | *Hughes et al., 2010* | NA | |
| Software, algorithm | ImageJ | NIH | https://imagej.nih.gov/ij/ | |
| Software, algorithm | ClockLab Software | Actimetrics (Wilmette, IL) | https://actimetrics.com/products/clocklab | |

## Fly husbandry and stocks

Fly stocks and crosses were maintained at room temperature or at 18°C on standard cornmeal molasses medium. Stocks for RNAi overexpression were obtained from VRDC. *brr2*[e03171], *prp8*[2e1], *prp8*[2e2] mutants were obtained from the Bloomington stock center. *iso*[31], *per*[01] and *Gal4* stocks were from the Sehgal lab stock collection. Transgenic lines for *tim* cDNA overexpression were generated by the site-specific PhiC31 Integration System (Rainbow Transgenics) using the attP on the 3[rd] chromosome. The DNA for fly embryo injections contained *tim* cDNA constructs (*Figure 5A*) subcloned into pUASTattB vectors.

## Circadian behavior analysis

For free-running circadian analysis, male flies were entrained to 12 hr:12 hr light:dark (LD) cycles at 25°C for at least three complete cycles. Flies were loaded into TriKinetics Drosophila Activity Monitor (DAM) system (Trikinetics, Waltham, MA), released into constant darkness and recorded for at least 7 days. Circadian parameters (period and rhythm strength) were determined using Clocklab Software (Actimetrics, Wilmette, IL). Period length was determined with $\chi^2$ periodogram analysis. Rhythm strength was determined using Fast Fourier Transform (FFT) values. A fly was considered rhythmic if the FFT value was greater than 0.01.

For analysis of circadian behavior in LD, flies were stably entrained in LD at 25°C for 3 days and their behavior was recorded as described above for the next three subsequent days. The analysis of activity counts was performed using Insomniac3 Software (RP Metrix).

## Immunohistochemistry and confocal microscopy

Fly brains were dissected in 4% paraformaldehyde (PFA) in 1x phosphate-buffered saline (PBS), fixed in PFA for 20 min at room temperature (RT) and then additionally trimmed of the air sacs and other contaminating tissues in 1xPBST. Dissected brains from each time point were stored in 1xPBST at

4°C until all of the time points were collected (never for more than 8 hr). Once all dissections and fixations were completed, brains were washed in 1xPBST for 20 min at RT, blocked with 5% Donkey Serum (DS) in 1xPBST for 20 min at RT, and incubated with the primary antibodies diluted in 5% DS overnight with gentle shaking at 4°C. The primary antibodies included rat anti-TIM (UPR42, 1:1000), guinea pig anti-PER (UP1140, 1:1000), rabbit anti-PDF (HH74, 1:500) and mouse anti-LaminC (LC28.26 from Developmental Studies Hybridoma Bank, 1:500). After a 30 min wash in 1xPBST at RT, brains were incubated with secondary antibodies diluted in 5% DS at RT. Secondary antibodies were used at 1:500 dilution and included FITC donkey-anti-guinea pig (Rockland), Alexa555 donkey-anti-rabbit (Jackson Immuno), Alexa647 donkey-anti-mouse (Jackson Immuno). Samples were washed for 30 min in 1xPBST at RT, and mounted in VectaShield. Slides were imaged with a Leica SP5 confocal microscope using a 40x oil-immersion objective and a 0.5 µM step size. The signal was adjusted to be not saturated as determined by QLUT parameters. After this original adjustment, all the settings were kept constant for a given experimental set. ImageJ software was used for analysis.

## Western blot analysis

Western blot assays with fly heads were performed as previously described (*Garbe et al., 2013*). Briefly, 7–10 fly heads per genotype/condition were lysed in 1x Passive Lysis Buffer (Promega), supplemented with protease and phosphatase inhibitors. For S2 cell extracts, 48 hr after transfection, cells were collected and lysed in the same buffer as described for fly heads. The following primary antibodies were used: anti-PER (UP1140, 1:1000), anti-TIM (UPR42, 1:1000) and anti-HSP70 (Sigma, 1:5000).

## Plasmids and S2 cell culture

For cell culture expression, *tim* cDNA was subcloned into pIZ-V5 plasmids. Using standard restriction enzyme cloning technique, the *tim-tiny* intron was subcloned into pIZ-V5 vectors carrying intronless *tim* cDNA from the pBluescript-*tim* plasmid. Mutagenesis of the 5' splice donor of *tim-tiny* intron site was performed with primers catalogued in the Key resources table using Q5 Site-Directed Mutagenesis Kit (NEB).

S2 cells were cultured in a standard Schneider medium (Invitrogen) supplemented with 10% FBS (Sigma). Transfection was done using an Effectene kit (Qiagen) according to the manufacturer's protocol.

## Quantitative RT-PCR

Flies were collected on dry ice at indicated time points and stored at −80°C until all of the time points were collected (within 24 hr). Fly heads were then collected on dry ice and homogenized with TRIzol (ThermoFischer) on ice using standard protocols. Following the phase separation, the aqueous phase was transferred into a new tube, mixed with an equal volume of 70% ethanol and loaded directly onto the RNeasy mini kit columns (Qiagen). The rest of the RNA isolation was done according to the manufacturer's protocol. On-column DNase digestion (Qiagen) for 15 min at RT was always included. cDNA was generated with Superscript II (Invitrogen), according to the manufacturer's protocol. Quantitative RT-PCR reaction was performed in a ViiA7 Real-Time PCR system (Applied Biosystems) using SYBR Green Master Mix (Applied Biosystems) with gene specific primers. Relative gene expression was calculated using the ΔΔCt method with *rp49* as normalization control.

## RNA-Sequencing and data analysis

RNA extraction was performed as described in the 'Quantitative RT-PCR' section above. Tapestation (Agilent) was used to ensure that all of RNA samples were of high-quality (RIN >8). five samples per each genotype were selected and prepared with Lexogen's SENSE mRNA-Seq library Prep Kit. Illumina Next-Generation Sequencing (NextSeq 500) with 300pb paired-end high output run (at ~50M reads per sample) was performed by the Genomics Facility at the Wistar Institute, Philadelphia, PA.

The RNA-seq reads were aligned to the Drosophila genome (dm6.BDGP6.v88) using STAR version 2.5.3a (*Dobin et al., 2013*). Normalization and quantification were performed with the PORT version 0.8.2a-beta pipeline (*Grant, 2018*) which first removes reads that map to ribosomal RNA sequences or mitochondrial DNA and then uses a read re-sampling strategy for normalization to minimize unwanted variance such as differences in sequencing depth among the samples. PORT

normalization was performed before quantification, at the aligned read level. After normalization, the quantification of features (genes, exons, introns, junctions) was done with respect to the Ensemblv88 annotation. The differential expression analysis was performed using the R Bioconductor package limma-voom (*Ritchie et al., 2015*). General pathway enrichment analyses were performed using DAVID (https://david.ncifcrf.gov/). The top 743 differentially expressed genes upon *prp4* downregulation, corresponding to FDR $\leq$ 0.05, were used for pathway enrichment analysis. Differential splicing analysis for *prp4* knockdown samples (with respect to the control) was performed using CASH (*Wu et al., 2018*). Full transcript quantification was done using Cufflinks-2.2, and differential psi (percent spliced in) analysis was performed for various gene isoforms to identify alternate splicing events (*Trapnell et al., 2013*; *Trapnell et al., 2010*).

### Quantification and statistical analysis

The statistical parameters are included in the legends of each figure. JTK_CYCLEv3.1 was run in R for circadian statistical analyses. GraphPad Prism was used for all other statistical tests.

### Data deposition

The RNA-Seq data generated in this work are freely available at the Gene Expression Omnibus (GEO) standard repository (accession # GSE115163).

## Acknowledgements

We thank David Garbe for his assistance with the original kinase screen. The work was supported by an NIH grant, R37NS048471, to AS and UL1TR001817 (Institutional Clinical and Translational Science Award), to Garret Fitzgerald.

## Additional information

### Funding

| Funder | Grant reference number | Author |
|---|---|---|
| NIH Clinical Center | UL1TR001817 | Gregory R Grant |
| National Institute of Neurological Disorders and Stroke | R37NS048471 | Amita Sehgal |

The funders had no role in study design, data collection and interpretation, or the decision to submit the work for publication.

### Author contributions

Iryna Shakhmantsir, Conceptualization, Data curation, Formal analysis, Validation, Investigation, Visualization, Methodology, Writing—original draft, Writing—review and editing; Soumyashant Nayak, Software, Formal analysis, Writing—original draft; Gregory R Grant, Data curation, Software, Formal analysis, Supervision, Methodology, Project administration, Writing—review and editing; Amita Sehgal, Conceptualization, Supervision, Funding acquisition, Project administration, Writing—review and editing

### Author ORCIDs

Iryna Shakhmantsir (ID) http://orcid.org/0000-0003-2138-0236
Amita Sehgal (ID) http://orcid.org/0000-0001-7354-9641

### Decision letter and Author response

Decision letter https://doi.org/10.7554/eLife.39821.023
Author response https://doi.org/10.7554/eLife.39821.024

# Additional files

## Supplementary files

• Supplementary file 1. DAVID pathway enrichment analysis of all differentially expressed genes upon *prp4* knockdown
DOI: https://doi.org/10.7554/eLife.39821.014

• Supplementary file 2. List of differentially expressed genes upon *prp4* knockdown
DOI: https://doi.org/10.7554/eLife.39821.015

• Supplementary file 3. Differentially spliced events upon *prp4* knockdown identified with CASH
DOI: https://doi.org/10.7554/eLife.39821.016

• Supplementary file 4. Differentially spliced isoforms upon *prp4* knockdown identified with Cufflinks-2.2 pipeline
DOI: https://doi.org/10.7554/eLife.39821.017

• Supplementary file 5. Calculation of the ratio of *tim-tiny* retaining isoforms to all other isoforms using Cufflinks-2.2 output
DOI: https://doi.org/10.7554/eLife.39821.018

• Transparent reporting form
DOI: https://doi.org/10.7554/eLife.39821.019

## Data availability

Sequencing data have been deposited in GEO under accession code GSE115163.

The following dataset was generated:

| Author(s) | Year | Dataset title | Dataset URL | Database and Identifier |
|---|---|---|---|---|
| Amita Sehgal, Iryna Shakhmantsir, Soumyashant Nayak, Gregory R. Grant | 2018 | RNAseq of *prp4* knockdown in Drosophila | http://www.ncbi.nlm.nih.gov/geo/query/acc.cgi?acc= GSE115163 | NCBI Gene Expression Omnibus, GSE115163 |

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
