## [Decision Letter]

[**Editorial note:** This article has been through an editorial process in which the authors decide how to respond to the issues raised during peer review. The Reviewing Editor's assessment is that all the issues have been addressed.]

Thank you for submitting your article "Spliceosome Factors Target *timeless (tim*) to Maintain Clock Function and Circadian Behavior in *Drosophila*" for consideration by *eLife*. Your article has been reviewed by three peer reviewers, and the evaluation has been overseen by a Reviewing Editor and K VijayRaghavan as the Senior Editor. The following individual involved in review of your submission has agreed to reveal his identity: John Ewer (Reviewer #1). The other reviewers remain anonymous.

The Reviewing Editor has highlighted the concerns that require revision and/or responses, and we have included the separate reviews below for your consideration. If you have any questions, please do not hesitate to contact us.

Summary:

As described in the authors' Abstract: "Transcription-translation feedback loops that comprise eukaryotic circadian clocks rely upon temporal delays that separate the phase of active transcription of clock genes, such as *Drosophila period (per)* and *timeless (tim),* from negative feedback by the two proteins. However, mechanisms underlying such delays are unclear. Through an RNA interference screen, [the authors] found that pre-mRNA processing 4 (PRP4) kinase, a component of the U4/U5.U6 triple small nuclear ribonucleoprotein(tri-snRNP) spliceosome, and other tri-snRNP components regulate cycling of the molecular clock as well as rest:activity rhythms. Unbiased RNA-Sequencing uncovered an alternatively spliced intron in *tim* whose increased retention upon *prp4* downregulation leads to decreased TIM levels. [They] demonstrate that the splicing of *tim* is rhythmic with a phase that parallels delayed accumulation of the protein in a 24-hour cycle. [The authors] propose that alternative splicing [(in addition to regulation of protein translation)] constitutes an important clock mechanism for delaying the daily accumulation of clock proteins, and thereby negative feedback by them."

Major concerns:

Please respond to all of the major comments from all three reviewers since they raise specific items that should be addressed. As pointed out by Reviewer 3, "the major unresolved conceptual issue is how the differential splicing of this intron is produced throughout the day by the clock. The authors speculate that daily changes in the level of splicing factors (and there is some evidence for this in earlier tissue-specific RNA profiling analyses) might produce the altered intron inclusion. Since *prp4* is a kinase, there might also be circadian regulation of its kinase activity. This possibility could be mentioned in the discussion.” The minor comments from Reviewer 3 should also be addressed.

As suggested by Reviewer 2, the title does not say much about the mechanism. The authors should consider the title suggested by Reviewer 2: "Spliceosome Factors Target timeless mRNA to control protein accumulation in the *Drosophila* circadian oscillator."

The original reviewers’ comments are appended below.

Separate reviews (please respond to each point):

*Reviewer #1:*

This is an interesting manuscript that shows that the regulation of the spliceosome itself contributes to the mechanisms that set the periodicity of the circadian clock. It nicely extends recent findings by Wang et al., 2018 (10.7554/*eLife*.35618) by identifying an intron in the timeless gene whose retention, due to actions mediated by components of the U4/U5.U6 spliceosome, (likely) delays the accumulation of TIM protein, thereby affecting the exact periodicity of the clock.

I am no expert in RNAseq analyses, so can't weigh in on this issue. The rest of the work is solid and well documented, and the conclusions are warranted. Some of the effects are quite modest, but then again, the effects on the clock, while biologically very significant, are "numerically" also modest (in few areas of biology is a change from ~24 to ~26 very impressive).

My only comments refer to the text.

1) Abstract "However, mechanisms underlying such delays are unclear." I would say that the mechanisms are not "unclear" but that "our understanding of the mechanisms involved is incomplete" since there is clear evidence (mentioned in the manuscript, for instance, in the Introduction final paragraph) that phosphorylation, protein stability, and dimerization of PER and TIM, all contribute to the functioning and period-setting of the clock.

2) Results section: "Overall, these findings strongly suggest that the free-running circadian component of the *prp4*-knockdown phenotype maps to LNvs." Rather than "maps" I would say, "the function of *prp4* in LNvs contributes to setting the periodicity of the clock". This because the knocking down *prp4* in all clock neurons (Figures 1A, B) is not quite the same as knocking it down only in LNvs (Figures 1D-G).

3) Discussion section: "Importantly, our findings address a longstanding question in the circadian field, specifically how negative feedback by clock proteins is delayed in order to permit distinct phases, and therefore oscillations, of transcriptional activation and repression." Rather than "address" I would say, "contribute to our understanding of".

4) In the Introduction and Discussion I think it is appropriate to cite the work of Beckwith et al., 2017 (10.1534/genetics.117.300139), which shows that alternative-splicing regulator, SR-related matrix protein of 160 kDa (SRm160), affects gene expression in pacemaker neurons.

5) In the Discussion I think it is worth mentioning another situation where a specific alternative splicing event has an impact on clock function, namely the BK potassium channel, where inclusion of a specific exon changes the electrophysiological properties of the channel, most likely contributing to the circadian changes in firing rate of SCN (work from Andrea Meredith's lab).

Minor Comments:

Difficult to resist suggesting that rather than "*tim-tiny*", this intron be called "Tiny tim"….

Additional data files and statistical comments:

The most relevant data (RNA-Seq data) have been made available (Gene Expression Omnibus standard repository, accession # GSE115163).

*Reviewer #2:*

This is a very interesting study by the Sehgal lab that reveals a new mechanism regulating TIM protein accumulation by producing a truncated unstable TIM protein during daytime. This is a new post-transcriptional mechanism that sheds light on the most obscure part of the negative feedback loop: how is PER/TIM protein accumulation delayed in comparison to the mRNA oscillation. The study represents a lot of smart work, the experiments are clearly presented and it is relatively easy to follow the paper. The data look really convincing although I will have a couple of suggestions.

– Results section and Figure 1: complex behavioral rhythms are obtained when *prp4* is downregulated in PDF cells whereas long period rhythms are obtained with all-clock neurons downregulation (Figure 1). To me, it suggests that *prp4* downregulation has also effects in pdf- oscillators, which is no surprise. I would not simply conclude that the *prp4*-knockdown phenotype maps to LNvs.

– Subsection “PRP4 is required for robust TIM and PER cycling” and Figure 2: the PER phosphorylation phenotype is strong (Figure 2D). The authors propose that it is a consequence of a defect in PER nuclear transfer, due to TIM decrease. I think that the possibility of a direct effect of *prp4* on PER should be tested by comparing PER phosphorylation in *prp4*+ and *prp4* RNAi flies with a *tim^0^* genetic background. I acknowledge that it might not be easy to see since TIM affects PER phosphorylation, but I believe that it is an important point to clarify.

Would it be possible to analyze the effects of a similar decrease of TIM (50% decrease at ZT18-20 based on Figure 2E) but as a consequence of another molecular defect, on PER phosphorylation?

– Subsection “PRP4 regulates *tim* splicing”: It is unclear to me how the 45 genes exhibiting differential splicing (Supplementary file 4, which I could not find) compare to the 5 genes that are described in Figure 3A. Is it the different analysis method? Please clarify.

– Subsection “PRP4 regulates tim splicing” third paragraph and Figure 3 (+ Figure 4—figure supplement 2): I understand that *tim-tiny* mRNA is increased by two-fold in *prp4* RNAi flies (Figure 3D). However, I suppose that the relative quantity of tim-tiny mRNA in comparison to total *tim* mRNA is a key parameter when it comes to explaining the effects of splicing regulation on total TIM protein levels. Please clearly indicate in the text what fraction of total *tim* mRNA *tim-tiny* represents in wild type flies.

It seems to me the effects of the different *tim* cDNAs on TIM protein levels (S2 cells and transgenic flies in Figure 4) are difficult to extrapolate to wild type and *prp4* RNAi flies if this ratio is not known. Does Figure 4—figure supplement 2B indicate that the spliced and retained forms are present in similar amounts?

Figure 5 and 6: I understand that the *tim-tiny/tim* ratio is indeed close to 1 in average. If I understand this well, that would say that half of *tim* transcripts do not produce a functional TIM protein. I believe that this point should be emphasized in the paper even before telling about the cycling of this ratio.

Minor Comments:

– The title does not tell much about the mechanism and I would suggest to make it more specific. For example, something in this line: Spliceosome Factors Target timeless mRNA to control protein accumulation in the *Drosophila* circadian oscillator

– Zheng and Sehgal, 2012 (Introduction section) does not seem to be in the list of references

– Figure 2 legend: E is quantification from D rather than A?

*Reviewer #3:*

This manuscript describes a well done investigation of broad interest to the circadian rhythm field because it demonstrates a novel clock-controlled splicing event in the RNA of a circadian clock gene (timeless, or *tim*) and makes a compelling argument that this splicing event introduces a delay in the accumulation of full length TIM protein. Such a delay has been postulated to be essential for the generation of oscillating expression for circadian proteins like PERIOD (PER) and TIM, but the delay has been proposed to arise from post-translational regulation of protein phosphorylation and stability (The post-translational effects are not disputed here; splicing is an additional regulatory step). Early circadian regulation of *per* and *tim* splicing has been shown to be needed for photoperiodic and temperature input pathways to the clock, but not for the underlying oscillation itself. Therefore, this manuscript will be of general interest. It is a very complete effort, with support for the role of RNA splicing coming from alterations to free-running period, rhythmicity and PER/TIM expression from knock-down of several splicing factors, identification in RNA-seq studies of a *tim* intron that is preferentially maintained with knock-down of one of these splicing factors (*prp4*), demonstration of altered effects on the oscillator with expression of mRNAs specific for the different alternative splice products, and demonstration of enhanced splicing for inclusion of the intron during the middle of the day or with higher temperatures, under conditions when it could delay the accumulation of TIM protein. The enhanced splicing event is particularly prominent in DD and may comprise a specific mechanism to confer phase delays of TIM accumulation in DD.

The major unresolved conceptual issue is how the differential splicing of this intron is produced throughout the day by the clock. The authors speculate that daily changes in the level of splicing factors (and there is some evidence for this in earlier tissue-specific RNA profiling analyses) might produce the altered intron inclusion. Since *prp4* is a kinase, there might also be circadian regulation of its kinase activity. This possibility could be mentioned in the Discussion.

Minor Comments:

1) Two lines on Table 1 are both labeled TUG>Dcr2;*prp4* RNAi (GD), but one of these (probably the second one) should perhaps be labeled KK instead of GD. Do these lines target different regions of the *prp4* mRNA to show the effects are target specific? And why does the "stronger" KK line have less effect on period length and rhythmicity than the GD line (Figure 1F and G)?

2) Define the differences between the boxes and error bars on Figure 1.

3) While the text states that the morning peak of activity is triggered by lights in Figure 1C, there still seems to be some anticipation – just less than in the control.

4) The complex rhythm in Figure 1D seems to include an ~6hr phase shift at the end of the record. Please comment on this in the text. The figure legend indicates that these rhythms are double-plotted but they appear to be single-plotted.

5) Why are the intervals for the collection of samples for TIM analysis in figure 2E 2hr long while they are precise times for PER analysis?

6) There does not seem to be a Supplementary file 3 nor a Supplementary file 4.

7) The model presented in Figure 6 argues that the delayed accumulation of TIM during the day due to inclusion of the extra intron (which truncates the reading frame of the protein) introduces a phase delay into the clock. But overexpression of cDNA with this intron missing produces longer periods, which apriori is inconsistent with the model (it should speed the clock, not slow it). Presumably the overexpression is producing some additional effects like prolonged repression of the E box, outside the confines of the model. Some discussion of this point is needed.

Additional data files and statistical comments:

Supplementary files 3 and 4 are missing and are important. They need to be provided. The statistical analysis is fine.

---

## [Author Response]

Reviewer #1:

[…] My only comments refer to the text.1) Abstract "However, mechanisms underlying such delays are unclear." I would say that the mechanisms are not "unclear" but that "our understanding of the mechanisms involved is incomplete" since there is clear evidence (mentioned in the manuscript, for instance, in the Introduction final paragraph) that phosphorylation, protein stability, and dimerization of PER and TIM, all contribute to the functioning and period-setting of the clock.2) Results section: "Overall, these findings strongly suggest that the free-running circadian component of the prp4-knockdown phenotype maps to LNvs." Rather than "maps" I would say, "the function of prp4 in LNvs contributes to setting the periodicity of the clock". This because the knocking down prp4 in all clock neurons (Figures 1A, B) is not quite the same as knocking it down only in LNvs (Figures 1D-G).3) Discussion section: "Importantly, our findings address a longstanding question in the circadian field, specifically how negative feedback by clock proteins is delayed in order to permit distinct phases, and therefore oscillations, of transcriptional activation and repression." Rather than "address" I would say, "contribute to our understanding of".

For points 1-3, the text of the manuscript was revised as suggested.

4) In the Introduction and Discussion I think it is appropriate to cite the work of Beckwith et al., 2017 (10.1534/genetics.117.300139), which shows that alternative-splicing regulator, SR-related matrix protein of 160 kDa (SRm160), affects gene expression in pacemaker neurons.

The following sentence was included in the Discussion: “In *Drosophila*, the alternative splicing regulator SR-related matrix protein of 160 kDa (SRm160) modulates PER levels locally in the pacemaker neurons to regulate circadian rhythmicity (Beckwith et al., 2017)”.

5) In the Discussion I think it is worth mentioning another situation where a specific alternative splicing event has an impact on clock function, namely the BK potassium channel, where inclusion of a specific exon changes the electrophysiological properties of the channel, most likely contributing to the circadian changes in firing rate of SCN (work from Andrea Meredith's lab).

The following sentence was included in the Discussion: “Alternatively, splicing mechanisms could regulate diurnal rhythmicity of neuronal excitability, as proposed for splicing of BK channels in the suprachiasmatic nucleus (Shelley et al., 2013).”

Minor Comments:Difficult to resist suggesting that rather than "tim-tiny", this intron be called "Tiny tim"….Additional data files and statistical comments:The most relevant data (RNA-Seq data) have been made available (Gene Expression Omnibus standard repository, accession # GSE115163).

Reviewer #2:

[…] The data look really convincing although I will have a couple of suggestions.– Results section and Figure 1: complex behavioral rhythms are obtained when prp4 is downregulated in PDF cells whereas long period rhythms are obtained with all-clock neurons downregulation (Figure 1). To me, it suggests that prp4 downregulation has also effects in pdf- oscillators, which is no surprise. I would not simply conclude that the prp4-knockdown phenotype maps to LNvs.

We agree that the circadian effects of *prp4* are unlikely to be mediated exclusively by sLNvs. However, because the *pdf-*selective downregulation of *prp4* can drive altered circadian behavior, we conclude that *pdf+* clock cells contribute to setting the periodicity of the clock. As per Reviewer 1’s suggestion, we modified the text to “Overall, these findings suggest that the function of PRP4 in LNvs contributes to setting the periodicity of the clock” instead of *"*Overall, these findings strongly suggest that the free-running circadian component of the *prp4*-knockdown phenotype maps to LNvs.”

Additionally, we modified the sentence to: “We identified PRP4 in a screen for novel regulators of the free-running circadian period, and established that PRP4 is necessary in clock *tim+* cells, to maintain 24-hour period and robust rhythmicity of the circadian clock.”

– Subsection “PRP4 is required for robust TIM and PER cycling” and Figure 2: the PER phosphorylation phenotype is strong (Figure 2D). The authors propose that it is a consequence of a defect in PER nuclear transfer, due to TIM decrease. I think that the possibility of a direct effect of prp4 on PER should be tested by comparing PER phosphorylation in prp4+ and prp4 RNAi flies with a tim^0^ genetic background. I acknowledge that it might not be easy to see since TIM affects PER phosphorylation, but I believe that it is an important point to clarify.Would it be possible to analyze the effects of a similar decrease of TIM (50% decrease at ZT18-20 based on Figure 2E) but as a consequence of another molecular defect, on PER phosphorylation?

This is a very good point. To better acknowledge that *prp4*-dependent PER phosphorylation phenotype could be independent of *prp4* effects on TIM, we added the following sentence to the Results section: “Alternatively, *prp4* depletion could have a TIM-independent effect on PER to regulate its phosphorylation profile.”

It would be very difficult to examine effects of *prp4* on PER phosphorylation in a *tim^0^* background because loss of TIM alone dramatically reduces levels of PER and affects PER phosphorylation, which would likely obscure effects of *prp4* (Price et al., 1995). We considered the possibility of examining a 50% decrease in TIM by using *tim^0^*/+ flies, but as these flies do not have a behavioral phenotype, they would not be relevant in terms of explaining the *prp4* phenotype. We believe that reduction of TIM in flies with *prp4* downregulation is greater in sLNvs than across the rest of the brain. Nevertheless, it is true that some aspects of the *prp4* phenotype might be independent of TIM, so it is best to acknowledge this as we propose to do above.

– Subsection “PRP4 regulates tim splicing”: It is unclear to me how the 45 genes exhibiting differential splicing (Supplementary file 4, which I could not find) compare to the 5 genes that are described in Figure 3A. Is it the different analysis method? Please clarify.

To clarify the methods used to analyze the RNA-Seq data, we changed the sentence to “Using the Comprehensive AS Hunting (CASH) method, which assays for splicing events, our analysis identified 45 genes exhibiting differential splicing upon *prp4* downregulation with FDR ≤ 0.05 (Wu et al., 2017) (Supplementary File 3).” To further narrow down relevant targets, we utilized additional algorithms and identified overlapping splicing events. We state this point as “Our initial splicing analysis was performed with CASH, but we additionally ran the Cufflinks-2.2 pipeline to obtain psi (percent spliced in) information for differentially spliced isoforms (Supplementary File 4).” The 5 genes described in Figure 4A are the overlapping targets identified by both of the splicing algorithms. We believe the legend of Figure 4 clearly makes this point: “(A) Only 5 genes were identified as differentially spliced upon *prp4* downregulation with both CASH (Wu et al., 2017) and Cufflinks/differential psi (percent spliced in) pipelines (Trapnell et al., 2010; Trapnell et al., 2013).”

– Subsection “PRP4 regulates tim splicing” third paragraph and Figure 3 (+ Figure 4—figure supplement 2): I understand that tim-tiny mRNA is increased by two-fold in prp4 RNAi flies (Figure 3D). However, I suppose that the relative quantity of tim-tiny mRNA in comparison to total tim mRNA is a key parameter when it comes to explaining the effects of splicing regulation on total TIM protein levels. Please clearly indicate in the text what fraction of total tim mRNA tim-tiny represents in wild type flies.It seems to me the effects of the different tim cDNAs on TIM protein levels (S2 cells and transgenic flies in Figure 4) are difficult to extrapolate to wild type and prp4 RNAi flies if this ratio is not known. Does Figure 4—figure supplement 2B indicate that the spliced and retained forms are present in similar amounts?Figure 5 and 6: I understand that the tim-tiny/tim ratio is indeed close to 1 in average. If I understand this well, that would say that half of tim transcripts do not produce a functional TIM protein. I believe that this point should be emphasized in the paper even before telling about the cycling of this ratio.

We agree that the ratio of isoforms that contain *tim-tiny (tim*-RM and *tim*-RS) to those that do not contain this intron is an important parameter to highlight in our study. We revisited our Cufflinks-2.2 analysis of the RNA-Seq data to better calculate this ratio (provided in the new Supplementary File 5). This analysis indicated that the isoforms that retain *tim-tiny* are twofold more abundant than isoforms that do not contain it (at ZT 8 when all of the RNA-Seq samples were collected). This finding would imply that at ZT8 the majority of *tim* isoforms do not produce full-sized TIM or produce unproductive protein that is rapidly degraded (Figure 5). As our data indicate, the ratio of spliced and unspliced forms changes over time (Figure 6A-B). The new information about the ratio is included in the following sentence: “To estimate how common *tim-tiny* retention was in control flies (*GMR/+)*, we quantified the ratio of isoforms that contain *tim-tiny (tim*-RM and *tim*-RS) to those that do not contain this intron using our Cufflinks-2.2 output (Supplementary File 4). This analysis indicated that the isoforms retaining *tim-tiny* were twice as abundant as other isoforms (at ZT 8 when all of the RNA-Seq samples were collected) (Supplementary File 5).”

Minor Comments:– The title does not tell much about the mechanism and I would suggest to make it more specific. For example, something in this line: Spliceosome Factors Target timeless mRNA to control protein accumulation in the Drosophila circadian oscillator.

To incorporate the effect on circadian behavior, we have changed the title to: “Spliceosome Factors Target timeless mRNA to control protein accumulation and circadian behavior in *Drosophila*”

– Zheng and Sehgal, 2012 (Introduction section) does not seem to be in the list of references

The reference was added.

– Figure 2 legend: E is quantification from D rather than A?

The figure legend was corrected.

Reviewer #3:

[…] The major unresolved conceptual issue is how the differential splicing of this intron is produced throughout the day by the clock. The authors speculate that daily changes in the level of splicing factors (and there is some evidence for this in earlier tissue-specific RNA profiling analyses) might produce the altered intron inclusion. Since prp4 is a kinase, there might also be circadian regulation of its kinase activity. This possibility could be mentioned in the Discussion.

The following sentence was added to the Discussion: “In addition to changes in total levels of PRP4, circadian regulation of its kinase activity could contribute to differential splicing of *tim* over the course of the day.”

Minor Comments:1) Two lines on Table 1 are both labeled TUG>Dcr2;prp4 RNAi (GD), but one of these (probably the second one) should perhaps be labeled KK instead of GD. Do these lines target different regions of the prp4 mRNA to show the effects are target specific? And why does the "stronger" KK line have less effect on period length and rhythmicity than the GD line (Figure 1F and G)?

The second line was corrected to “*TUG>Dcr2;prp4 RNAi (KK)”* in Table 1.

With respect to the strength of the KK line, we believe the reviewer is referring to the non-significant effect of KK on period with the *pdf* driver. However, please note that a very small percentage of these flies have a single rhythm that could be analyzed for period, as opposed to flies with GD driven by *pdf* (Figure 1E-F). We consider the KK line ‘stronger’ than GD because KK has a longer average period and less rhythmicity than GD when driven with the TUG driver (Figure 1B, Table 1).

The GD line (VDRC:27808) and KK line (VDRC:107042) target distinct, but overlapping regions of *prp4*.

2) Define the differences between the boxes and error bars on Figure 1.

The boxes extend from the 25th to 75th percentiles. The line in the middle of the box is plotted at the median. Whiskers extend from the lowest to highest value. Since the same description would be relevant for Figure 5D, we updated the legends of both Figure 1 and Figure 5.

3) While the text states that the morning peak of activity is triggered by lights in Figure 1C, there still seems to be some anticipation – just less than in the control.

The text was adjusted to “Compared to controls, the flies with circadian-cell-specific *prp4* knockdown (*TUG; Dcr2>prp4^RNAi^ (GD)*) exhibited a delayed evening activity peak as well as a slightly delayed morning peak”

4) The complex rhythm in Figure 1D seems to include an ~6hr phase shift at the end of the record. Please comment on this in the text. The figure legend indicates that these rhythms are double-plotted but they appear to be single-plotted.

The reviewer is correct in that there appears to be a phase shift (delay or advance) in many records. We have modified the sentence to read: “These complex rhythms were generally characterized by changes in the behavioral pattern, often manifest as phase shifts during day 4 or 5 of constant darkness (see ~6-h shift in the record shown), in the midst of an otherwise rhythmic record (Figure 1D).”

We apologize for the inaccuracy and have deleted the word ‘double-plotted’.

5) Why are the intervals for the collection of samples for TIM analysis in Figure 2E 2hr long while they are precise times for PER analysis?

There was no strategic reason for this. For the TIM analysis, we pooled data from multiple experiments in which samples were collected at similar, but not identical, times so we indicated the collection times as intervals.

6) There does not seem to be a Supplementary file 3 nor a Supplementary file 4.

These files will be uploaded with the final paper submission.

7) The model presented in Figure 6 argues that the delayed accumulation of TIM during the day due to inclusion of the extra intron (which truncates the reading frame of the protein) introduces a phase delay into the clock. But overexpression of cDNA with this intron missing produces longer periods, which apriori is inconsistent with the model (it should speed the clock, not slow it). Presumably the overexpression is producing some additional effects like prolonged repression of the E box, outside the confines of the model. Some discussion of this point is needed.

Overexpression of *tim* cDNA from the UAS promoter typically results in longer periods as it does not allow for feedback and so leads to massive accumulation of TIM (Yang and Sehgal, 2001). Please note that earlier studies showing period-shortening with PER over-expression used constructs that included a *per* promoter that could be repressed by feedback (discussed in Yang and Sehgal, 2001). It is true that we do not see a period difference with ‘*tim-*retained’ and ‘*tim-*spliced’ overexpression constructs (Figure 5D), which may be because these constructs have small effects in a wild type background. However, the reviewer is absolutely correct in that the “intron missing” form should produce shorter periods than the ‘*tim-*retained’ form, and we now show the data for expression of these constructs in a *tim^0^* background (included as Figure 5E-F; also shown below). Note that both forms rescue with a longer period, as expected of UAS constructs, but rescue with the spliced form (‘*tim-*spliced’) yields a shorter period, albeit in a smaller percentage of flies. The latter point of lower rhythmicity also supports our idea that delays are required for a robust oscillator.

We added the following sentences to the legend of Figure 5: “(E) *TUG*-driven expression of *tim* cDNA, with both ‘*tim*-spliced’ and ‘*tim*-retained’ constructs, rescues *tim^0^* circadian rhythmicity. n.s., not significant at the 0.05 level, **p ≤ 0.01, ****p ≤ 0.0001 by pairwise Fischer’s exact test (n=28-41). (F) *TUG*-driven rescue of *tim^0^* circadian rhythms with *tim* cDNA that includes *tim-tiny (tim^0^,TUG>tim*-retained) lengthened the period compared to the rescue with *tim* cDNA lacking *tim-tiny (tim^0^,TUG>tim*-spliced). ***p ≤ 0.001 by Student’s t test, n=10-22.”

These findings were summarized in the Results section as “To further understand the effect of *tim-tiny* splicing on circadian behavior, we overexpressed ‘*tim*-spliced’, ‘*tim*-retained’ and ‘*tim*-retained-ssM’ using the *TUG* driver in the *tim^0^*homozygous background. We hypothesized that splicing of *tim-tiny* is necessary for the maintenance of circadian rhythmicity and so the ‘*tim*-retained’ construct would rescue the behavioral rhythmicity most efficiently because it allows for both the splicing and retention of the intron. Additionally, because *prp4* knockdown increases the retention of *tim-tiny* (Figure 4) and prolongs the rhythm (Figure 1), we speculated that the ‘*tim*-spliced’ construct would rescue with a shorter period than the ‘*tim*-retained’ construct. As expected, the expression of *tim* cDNA with a 5’ splice site mutation in *tim-tiny* (‘*tim-*retained+ssM’) did not restore rhythms in *tim^0^* flies (Figure 5E). The other two cDNA constructs, ‘*tim*-retained’ and ‘*tim*-spliced’ rescued the rhythmicity in 54% and 32% of *tim^0^* flies, respectively. Importantly, there was a significant difference in period length between the flies that were rhythmic (Figure 5F). As discussed above, the UAS-GAL4 system typically over-expresses proteins and so rescues *per/tim* mutants with longer periods (Yang and Sehgal, 2001), which we observed for the ‘*tim*-retained’ isoform (~26 hours). Shorter periods were seen with the ‘*tim*-spliced’ version (Figure 5F), supporting the idea that *tim-tiny* retention promotes clock delays.”

We changed the title of Figure 5 to “Intron retention in *tim* decreases TIM levels and affects circadian behavior” to better convey the message of the new panels.

Additional data files and statistical comments:Supplementary files 3 and 4 are missing and are important. They need to be provided. The statistical analysis is fine.

The files will be uploaded.